# Inheritance of associative memories and acquired cellular changes in *C. elegans*

Noa Deshe[1], Yifat Eliezer[1], Lihi Hoch[1], Eyal Itskovits [1], Eduard Bokman[1], Shachaf Ben-Ezra [1] & Alon Zaslaver [1] ✉

Experiences have been shown to modulate behavior and physiology of future generations in some contexts, but there is limited evidence for inheritance of associative memory in different species. Here, we trained *C. elegans* nematodes to associate an attractive odorant with stressful starvation conditions and revealed that this associative memory was transmitted to the F1 progeny who showed odor-evoked avoidance behavior. Moreover, the F1 and the F2 descendants of trained animals exhibited odor-evoked cellular stress responses, manifested by the translocation of DAF-16/FOXO to cells' nuclei. Sperm, but not oocytes, transmitted these odor-evoked cellular stress responses which involved H3K9 and H3K36 methylations, the small RNA pathway machinery, and intact neuropeptide secretion. Activation of a single chemosensory neuron sufficed to induce a serotonin-mediated systemic stress response in both the parental trained generation and in its progeny. Moreover, inheritance of the cellular stress responses increased survival chances of the progeny as exposure to the training odorant allowed the animals to prepare in advance for an impending adversity. These findings suggest that in *C. elegans* associative memories and cellular changes may be transferred across generations.

The capacity to form memories allows individuals to make educated decisions based on past experiences. A key memory paradigm is known as the associative memory, where a link between two seemingly unrelated cues is established. A classic example is the Pavlovian dogs who associated food (the unconditioned stimulus, US) with an auditory cue (conditioned stimulus, CS). Consequently, when exposed to this sound cue, these dogs started salivating in expectation for the associated food[1].

Formation of associative memories becomes particularly advantageous when the associated US is unfavorable (*e.g.*, a shock or a stress) since encountering the CS induces memory retrieval and individuals may consequently anticipate the impending adversity and prepare for it in advance[2–5]. A compelling idea is that these valuable experiences may be epigenetically transmitted to subsequent generations and provide the descendent individuals with a fitness advantage upon recurrent exposure.

A rich body of literature describes how extreme life experiences modulate physiology and behavior of subsequent generations[6–13]. However, evidence for inheritance of associative memories is scarce: In rodents, mice trained to associate an odor with an aversive electric shock transferred the acquired memory to their descendants[14,15], and memory traces were evident even in the F3 generation, indicating of transgenerational, rather than intergenerational, inheritance mechanisms[16–18]. In *Caenorhabditis elegans*, repeated olfactory imprinting for at least four generations was stably inherited through multiple successive generations[19]. Furthermore, worms grown on pathogenic *Pseudomonas* learn to avoid the bacteria and pass this capacity to their offspring[20,21]. Remarkably, an RNA originating from the *Pseudomonas* bacteria induces avoidance behavior up to four generations downstream of the parental generation that was exposed to these bacteria[22,23].

---

[1]Department of Genetics, Silberman Institute of Life Science, Edmond J. Safra Campus, The Hebrew University of Jerusalem, Jerusalem 9190401, Israel. ✉e-mail: alonzas@mail.huji.ac.il

In that respect, *C. elegans* offer a powerful model to studying inheritance of associative memories and associated cellular changes: Their short 3-day life cycle enables rapid interrogation of multiple consecutive generations and their compatibility with a myriad of genetic manipulations already extracted evolutionarily conserved mechanisms that promote epigenetic inheritance, including neuropeptides, proteins, chromatin modifications, small RNAs and other factors[18,24–32]. Crucially, recent studies, primarily in *C. elegans*, demonstrated that the famous 'Weismann barrier' is breached and information from somatic neurons is transmitted to the germline to affect the physiology and behavior of subsequent generations[8,21,23,33–35].

Furthermore, equipped with a compact neural network consisting of 302 neurons[36–38], *C. elegans* worms form various types of learning[39–41], and the stereotypic animal-to-animal anatomy makes them an appealing system for mapping individual memory-storing neurons[3,41,42].

Here we show that associative memories and acquired cellular changes are inheritable in *C. elegans* worms: F1 progeny of animals trained to associate an attractive odorant with a stressful starvation showed odor-evoked avoidance, and both the F1 and the F2 descendants of the trained animals exhibited odor-evoked cellular stress responses. While shared epigenetic pathways (H3K9 and H3K36 methylation, the small RNA binding Argonaute NRDE-3, and neuropeptide secretion) underlie these two inheritance capacities, these odor-evoked outputs are seemingly uncoupled. Notably, the sperm, but not the oocytes, transmit the acquired cellular changes and the same chemosensory neuron that stored the information in the trained parental generation also carried the information in the descendants, where its sole activation sufficed to induce a systemic stress response. Together, these findings establish that associative memories and associative-based cellular changes may be transferred across generations.

## Results
### Associative aversive memories are heritable
We trained hermaphroditic worms to form an associative memory by coupling the favorable odorant isoamyl alcohol (IAA) with starvation (Fig. 1A and "Methods"). In particular, we used a spaced-training paradigm as this was demonstrated to form robust associative memories in *C. elegans*[43]. As a control, we included mock-trained animals which underwent the same starvation and recovery cycles, though they were never exposed to the odorant IAA. To test whether animals successfully associated IAA with the stressful starvation, we quantified animal's avoidance response, scored as the animal's capacity to switch from forward to backward locomotion upon exposure to the stimulus (Methods). The parental (P0) trained generation worms showed significant increased avoidance behavior when compared to the mock-trained animals, indicating that animals robustly associated IAA with the negative experience (Fig. 1B).

When assaying all F1 progeny, we could not detect an elevated avoidance response upon exposure to IAA (Fig. 1B). However, when assaying F1s selected from P0s that avoided the IAA, these animals showed a significant avoidance when compared to their respective F1s that originated from mock-trained animals (also pre-selected from avoiding P0s). Notably, the F1 generation of trained and mock-trained animals were never starved, nor were they exposed to the odorant IAA (until the avoidance assay). F2-generation animals did not show an avoidance response (compared to mock controls) following exposure to the CS IAA, even when assaying F2s originating from F1s that exhibited the avoidance behavior (Fig. 1B).

### Associative memories are rather stable
Next, we studied the stability of these memories in both the P0 and the F1 generations. For this, we analyzed if these memories could undergo extinction by cultivating the trained (and mock trained) animals with the CS IAA in the presence of food. Memories within the trained P0

animals underwent extinction only after three such cultivation cycles (Fig. 1C). After each cultivation cycle, we set aside avoiding animals and allowed them to lay the F1 progeny. Strikingly, each of the tested F1-generation groups showed robust memory capacities. Even progeny of the P0 animals, in which memory did eventually undergo extinction following the third cultivation cycle, still retained the memory (Fig. 1C). These results indicate that the associative aversive memory is rather stable and does not readily undergo extinction. Moreover, even when the memory undergoes extinction in the P0-trained animals, the information can still be transmitted to the progeny.

### H3K9/H3K36-me, sRNAs, and neuropeptides are involved in memory inheritance
We next studied which genes and pathways may be involved in transmitting the associative memories to the F1 progeny. In *C. elegans*, small RNAs and histone modifications are the best characterized mechanisms underlying epigenetic inheritance[18,24,26–28,30]. We therefore analyzed the capacity to inherit associative memories in mutant animals, defective in major genes involved in these pathways (Fig. 2).

First, we studied key histone modulators, previously implicated in mediating epigenetic inheritance in *C. elegans* animals[25,44–49], and quantified the capacity of animals, defective in these genes, to transmit the associative memory to their progeny. F1 mutants, defective in H3K9 mono/di-methylation (*met-2*) and tri-methylation (*set-25*), failed to inherit the memory from their trained P0 mutant parents who showed intact (WT level) capacities to avoid the conditioned odorant. Similarly, F1 mutants, defective in H3K36 methylation (*met-1*) also failed to inherit the memory, though their trained parents showed only a slight behavioral change compared to their mock-trained counterparts (Fig. 2A). WDR-5.1, part of the H3K4 methylation complex, and the heterochromatin protein HPL-2, which binds methylated histones, do not seem to play a major role in memory inheritance as F1 descendants show memory retrieval capacities, though avoidance of *wdr-5.1* F1 mutants is less efficient than that of their P0-trained parental generation (Fig. 2A).

Next, we analyzed key genes in the small RNA pathways that were shown to be involved in *C. elegans* transgenerational inheritance[26,32,50–55]. Trained P0 mutants, defective in *nrde-3*, an Argonaute responsible for transferring small RNAs from the cytoplasm to the nucleus in somatic cells[56], and mutants in the *hrde-1* gene, a key nuclear Argonaute for transgenerational inheritance that is expressed in germ cells[51], showed intact odor-induced avoidance outputs. However, their progeny failed to show these memory-evoked outputs (Fig. 2B), suggesting the involvement of these genes in memory inheritance. Mutants defective in *nrde-2*, the nuclear dsRNA-induced RNAi factor responsible for maintenance of small RNAs transgenerational inheritance[53,56], were able to retrieve the memory in both the P0 and the F1 generations (Fig. 2B), suggesting that this gene is not involved in memory formation and inheritance.

As neuropeptides are known to relay nutritional status to the germline[12,57,58], these signals may also play a role in transmitting the associative memories. We therefore analyzed memory inheritance in *egl-3* mutants that are defective in the neuropeptide secretion pathway[59,60]. While the parental trained generation showed intact odor-induced avoidance behavior, F1 worms lacked these memory-induced behavioral outputs, suggesting that neuropeptides may be involved in memory transmission (Fig. 2B). Together, these results place H3K9/H3K36 methylations, the Argonautes NRDE-3 and HRDE-1, and neuropeptides as potential players that mediate the inheritance of associative memories.

### Acquired cellular changes are heritable
Coupling starvation with IAA also leads to cellular and physiological changes such that subsequent exposure of the trained animals to IAA induces a fast systemic stress response[3]. This stress response is

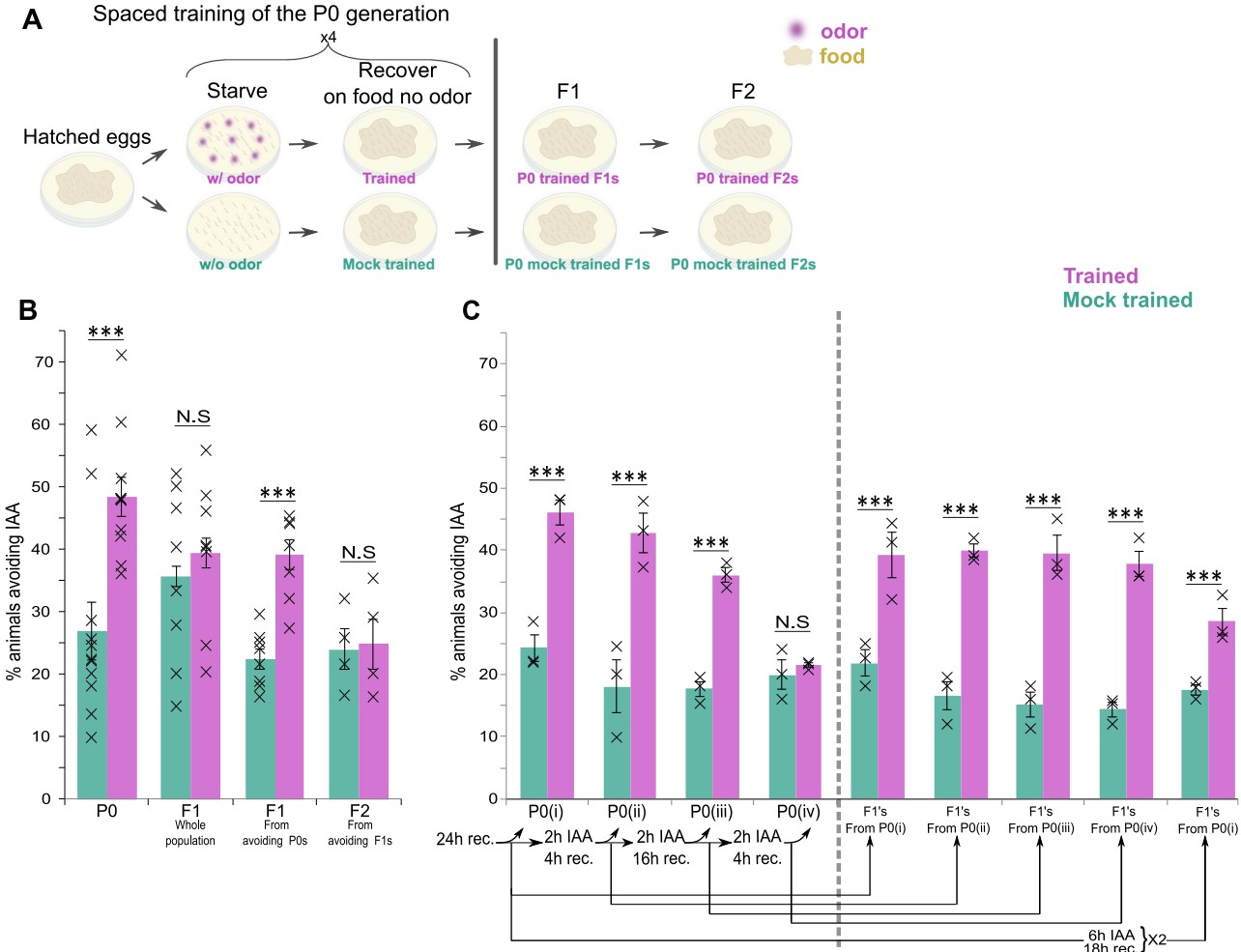

**Fig. 1 | The associative aversive memory is heritable and does not readily undergo extinction in both the P0 and the F1 generations. A** A spaced-training paradigm was used to form the aversive associative memory: P0-generation worms were intermittently starved four times in the presence (trained) or the absence (mock trained) of the conditioned odorant IAA. A full recovery on food in the absence of IAA followed each of the starvation steps. Subsequent generations, F1 and F2, were grown in normal satiety conditions. **B** Trained P0 animals and their F1 descendants showed avoidance behavior when presented with the CS IAA. Forward-moving animals were scored as avoiding if they stopped and backed within 3 seconds following the IAA presentation. Shown are the mean ± SEM of $N = 4$–9 biologically independent experimental repeats, each scoring ~50 animals. P-values from left to right: 5.3E−5, 0.23, 1.7E−4, 0.25. ***$p < 0.0001$. NS, not significant (two-sided proportion test, after Bonferroni correction). **C** Memory extinction attempts were performed by cultivating the animals with the

conditioned stimulus IAA in the presence of food followed by a recovery period (with food and in the absence of IAA). (i) Trained animals, 24 h post recovery from training, showed significant avoidance following odor-evoked memory reactivation. Trained P0 animals, exposed to IAA once (ii) or twice (iii) in the presence of food still showed significant odor-evoked responses. Lack of memory-induced avoidance was observed only after the third exposure to IAA+food (iv). F1 progeny of the P0 generation that underwent one (ii), two (iii), or three (iv) memory extinction attempts still showed robust memory traces. Memory did not undergo extinction after two long IAA exposures of the F1 progeny. Shown are the mean ± SEM of $N = 3$ biologically independent experimental repeats, each scoring ~50 animals. P-values from left to right: 8.3E−5, 5.4E−5, 0.5E−3, 0.87, 1.6E−3, 1.1E−5, 2.6E−5, 5.3E−5, 0.023 ***$p < 0.0001$ (two-sided proportion test, after Bonferroni correction). Significance comparisons are between trained and mock-trained animals. Source data is provided as a Source Data file.

manifested by the rapid translocation of the general stress response factor DAF-16/FOXO to nuclei of the gonad sheath cells. Anatomically, these cells are intimately associated with the germ cells[61], thus raising the possibility that the stressful information and the subsequent cellular changes may be transmitted to the progeny as well.

We therefore asked whether such cellular changes are also evident in the progeny of the trained hermaphrodites. For this, we trained animals as described above (Fig. 1A), and following recovery, we exposed them to IAA to score the translocation of DAF-16/FOXO to the cells' nuclei. Indeed, exposure to the CS IAA induced a rapid translocation of DAF-16/FOXO to cells' nuclei, indicating that the trained P0-generation worms formed robust cellular changes (Fig. 3A, C, and Supplementary Fig. 1). Interestingly, F1 descendants of these trained P0 hermaphrodites inherited these cellular changes, as exposing them to IAA also induced a rapid translocation of the DAF-16/FOXO to their

cells nuclei, despite the fact that this generation was never exposed to IAA, nor to starvation conditions (Fig. 3B, C, and Supplementary Fig. 1). Importantly, worms trained to couple IAA with food did not exhibit odor-evoked DAF-16/FOXO translocation, nor did their F1 progeny (Supplementary Fig. 2A). These starvation steps did not increase expression levels of DAF-16 in the trained parental generation, nor in their F1 progeny, thus precluding the possibility that animals undergoing repeated starvations had a lower threshold for initiation of the stress response due to higher levels of DAF-16 (Supplementary Fig. 2B). Furthermore, coupling IAA with starvation in the L1 or the L3 larval stages exclusively did not lead to cellular changes in the P0, nor in their F1 progeny (Supplementary Fig. 3). Training the P0 generation exclusively during the L4 stage formed robust cellular changes in the P0 animals, in agreement with a previous report[3], but these changes were not evident in the F1 progeny (Supplementary Fig. 3).

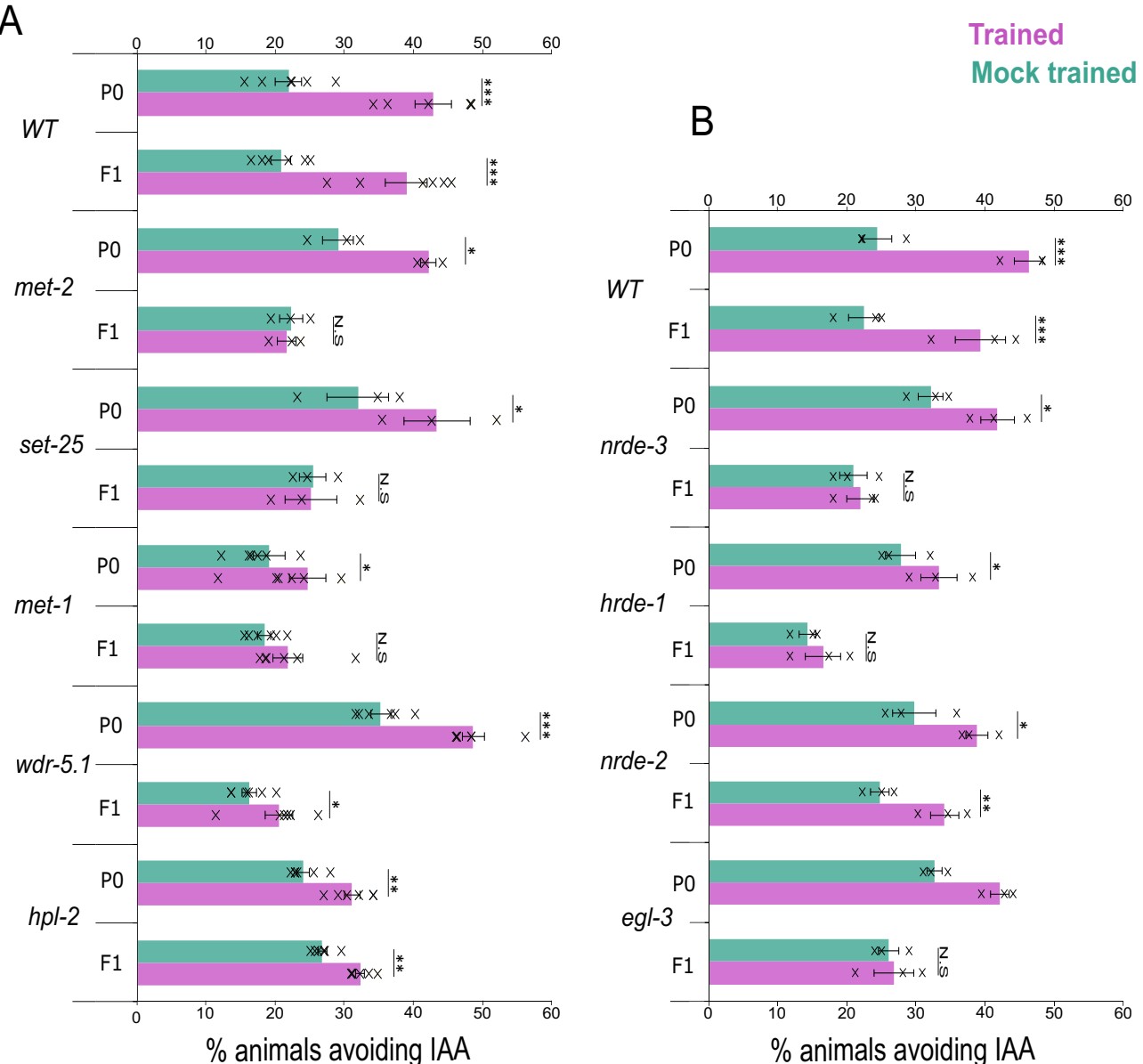

**Fig. 2 | Histone modifications, small RNA pathways, and neuropeptide secretion are involved in the inheritance of the associative aversive memory.** Quantification of the avoidance capacity following exposure to the CS IAA and memory reactivation in mutants, defective in histone modification genes (**A**) and in small RNA and neuropeptide secretion pathways (**B**). F1 progeny were picked from P0 animals that avoided IAA. For each group, shown are the independent experimental repeats (*N* = 3–6), each scoring ~50 animals. Significance comparisons are between trained and mock-trained animals. *P*-values from top to bottom: **A** 7.1E−6, 1.2E−3, 0.01, 0.4, 0.01, 0.47, 9.4E−3, 0.05, 2.8E−4, 0.01, 8.2E−4, 1.9E−3. **B** 4.3E−4, 5.5E−3, 0.02, 0.41, 0.03, 0.29, 0.02, 3E−3, 9.5E−3, 0.43. *p < 0.05, ***p < 0.0005 (two-sided proportion test, after Bonferroni correction). N.S not significant. Error bars denote SEM. Source data is provided as a Source Data file.

The cellular changes were also transmitted to the F2-generation animals, who similarly showed rapid DAF-16/FOXO translocation to cells' nuclei upon exposure to the CS IAA (Fig. 3C and Supplementary Fig. 1). This transmission was observed only in F2s that originated from F1s who showed IAA-induced DAF-16/FOXO nuclear translocation. F3-generation animals did not exhibit DAF-16/FOXO nuclear translocation following exposure to IAA, even when selecting F2s that showed the cellular response and assaying their progeny (Fig. 3C), indicating that this mode of inheritance was limited to two generations only.

**The inherited cellular changes and the associative memories are independent**

We next asked whether the inherited associative memory and the acquired cellular changes are coupled: that is, whether trained animals that show odor-induced avoidance will also exhibit odor-induced DAF-16/FOXO nuclear translocation (Fig. 3D–F). For this, we assayed trained animals for odor-induced avoidance, and segregated the assayed animals into avoiding and non-avoiding groups (Fig. 3E). Next, we assayed the animals from each group for odor-induced DAF-16/FOXO nuclear translocation. In both groups, a similar fraction of the animals exhibited odor-evoked nuclear translocation of DAF-16/FOXO (Fig. 3F), suggesting that the odor-evoked avoidance response and the odor-evoked cellular changes may represent two uncoupled processes.

**Acquired cellular changes are rather stable**

As the two processes are presumably decoupled, we also analyzed the stability of the acquired cellular changes. For this, we cultivated the trained animals with IAA while on food before re-challenging them with IAA to test DAF-16/FOXO nuclear translocation (similar to the

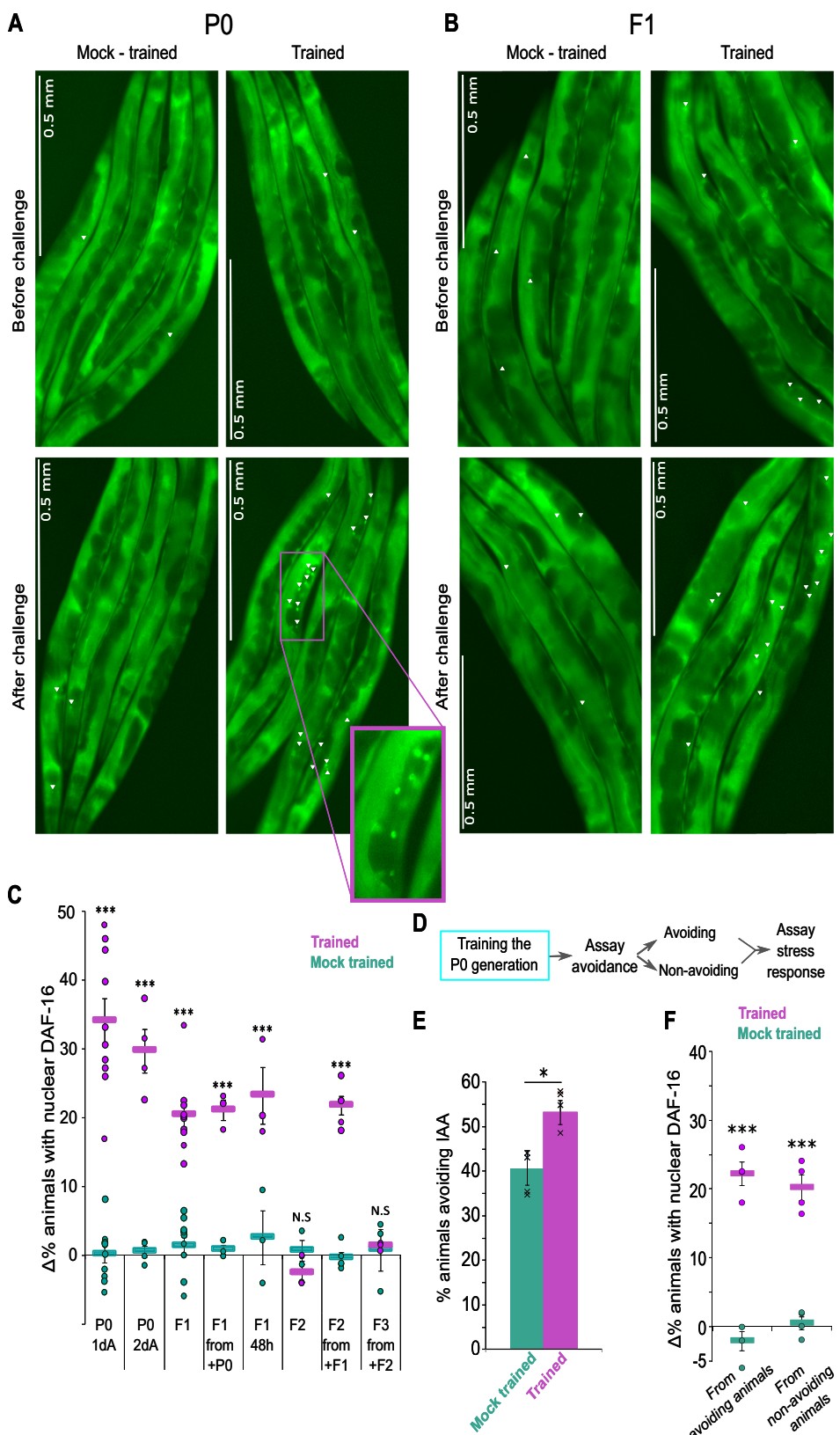

memory extinction experiments presented in Fig. 1C). Trained P0 animals, challenged with IAA following one or two cultivation cycles, showed robust odor-evoked DAF-16/FOXO nuclear translocation (Fig. 4). Only after the third cultivation cycle, the acquired cellular changes were not evident any more as a subsequent exposure to IAA did not induce the DAF-16/FOXO nuclear translocation.

Interestingly, F1 descendants of P0-trained animals, that were cultivated on food in the presence of IAA, either twice or three times, still exhibited odor-induced nuclear translocation of DAF-16/FOXO (Fig. 4). The inherited cellular changes in the F1 generation could be eliminated only after two cycles of cultivating the F1 animals on food in the presence of IAA (Fig. 4). Thus, even after the acquired changes in

**Fig. 3 | Acquired cellular changes are transmitted up to the F2 generation, and are presumably decoupled from the memory-evoked avoidance behavior.**
**A**, **B** Odor challenge induced a rapid (<20 min) translocation of DAF-16/FOXO to cells' nuclei. Shown are representative images of trained and mock-trained P0 (**A**) and F1 (**B**) animals before and after exposure to IAA (challenge). Nuclear transmitcation of DAF-16/FOXO was observed primarily in the gonad sheath cells (white arrowheads) and was visualized using a translational fusion of the protein to GFP[83]. We scored animals as initiating a stress response if at least six cells (in both gonad arms) showed nuclear DAF-16/FOXO localization. Mock-trained animals typically had ≤2 cells with nuclear DAF-16/FOXO (an extensive statistical validation for this scoring approach is found in[3]). Inset is a zoom-in image demonstrating the nuclear localization of DAF-16. Note images were cropped to allow zooming into the gonad sheath cells. **C** Quantification of the stress response based on the percent of animals detected with nuclear DAF-16/FOXO following exposure to training odorant IAA. Trained and mock-trained animal groups were each scored before and after challenging the animals with IAA. % of worms with nuclear DAF-16/FOXO before the challenge were subtracted from the % of worms with nuclear DAF-16/FOXO after the challenge (thus, presented as Δ%). Negative Δ% values could arise if % of animals with nuclear DAF-16/FOXO following the IAA challenge were lower than the % before the challenge (see Supplementary Fig. 1 for extended description and the raw data used to derive these values). +P0/+F1/+F2 denote that the assayed animals were descendants of animals in which the stress response was induced. 1dA and 2dA are 1- and 2-day-old adults, accordingly. 48 h denotes that the assayed F1s hatched 48 h post the recovery of the trained P0 animals. Otherwise, assays were

performed on F1s that hatched 24 h following recovery of the P0-trained animals, thus, F1s were not exposed to the training conditions in their embryonic stage. Shown are the mean ± SEM of N = 3–10 biologically independent experimental repeats, each scoring >50 animals. Significance comparisons are between trained and mock-trained animals. P-values from left to right: 3.8E−20, 2.2E−16, 3.2E−13, 2.5E−14, 2.2E−7, 0.19, 6E−13, 0.9. ***p < 0.0005 (two-sided proportion test, after Bonferroni correction). **D** Experimental design to test the coupling between behavioral avoidance and stress induction following exposure to the training odorant IAA. Trained and mock-trained P0 animals were first assayed for their odor-evoked avoidance capacity. Avoiding and non-avoiding animals were separately grouped, and each group was subsequently assayed for odor-evoked stress induction as manifested by DAF-16/FOXO nuclear translocation. For this we used the strain CF1934[86]. **E** Fraction of trained and mock-trained CF1934 animals that avoided IAA. Shown is the mean ± SEM of four independent experimental repeats, where each repeat consists of >50 animals from each group of animals. *p = 0.01 (two-sided proportion test after Bonferroni correction). Avoiding and non-avoiding animals from trained and mock-trained animals were then separately grouped. **F** Analysis of stress induction in the avoiding and the non-avoiding groups of both trained and mock-trained animals as collected following the behavioral assays shown in (**E**). Shown are the mean ± SEM of four independent experimental repeats, each scoring >50. Significance comparisons are between trained and mock-trained animals. P-values from left to right: 2.4E−7, 1.38E−5. ***p < 0.0005 (proportion test, two-sided, after Bonferroni correction). Source data is provided as a Source Data file.

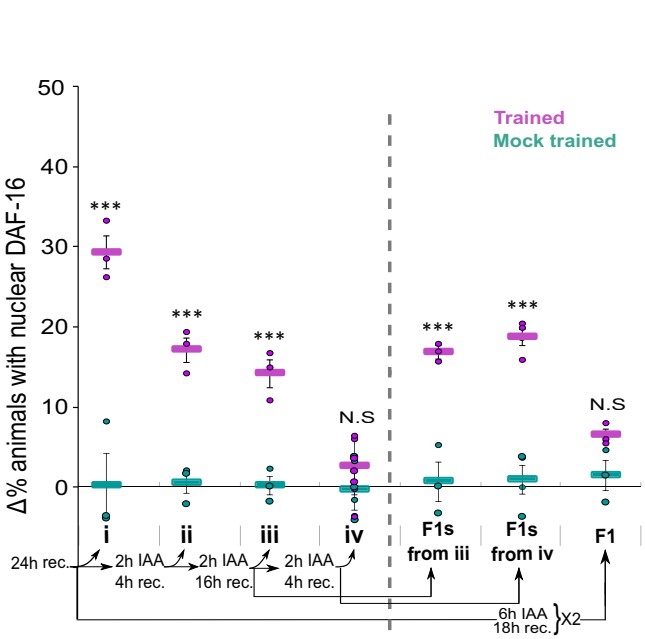

**Fig. 4 | The acquired cellular changes are rather stable and can be transmitted even when becoming undetectable in the P0-trained generation.** Trained P0-generation animals and their progeny were repeatedly cultivated with the CS IAA in the presence of food. Each such cultivation cycle was followed by a recovery period in the absence of IAA. (i) P0-generation animals showed significant IAA-induced nuclear translocation of DAF-16/FOXO 24 h post training. Trained P0 animals, exposed to IAA once (ii) or twice (iii) in the presence of food, still showed significant IAA-induced responses. These acquired changes were not evident following the third cultivation cycle (iv). F1 progeny of the trained P0 generation, that underwent two (from group iii) or three (from group iv) cultivation cycles of IAA+food, still showed robust IAA-induced responses. F1 animals that underwent two IAA+food cultivation cycles lost the IAA-induced responses. Shown are the means ± SEM of N = 3–7 biologically independent experimental repeats, each scoring ~50 animals. P-values from left to right: 6.7E−11, 6.6E−5, 2.2E−3, 0.18, 1.1E−4, 6.11E−8, 0.2. ***p < 0.0001 (proportion test, two-sided, Bonferroni correction). Significance comparisons are between trained and mock-trained animals. Source data is provided as a Source Data file.

the trained parental generation were lost, this information could still be successfully transmitted to the progeny.

### Acquired cellular changes increase survival chances
Odor-induced translocation of DAF-16/FOXO to cells' nuclei triggers rapid protective responses which enable animals to prepare in advance for an imminent adversity[3]. This odor-induced physiological response was indeed demonstrated to increase survival chances if animals subsequently encountered a stress[3]. We therefore asked whether the inherited cellular changes also endow the progeny with such protective capacities.

For this, we exposed trained and mock-trained worms of both the parental generation and their progeny to the CS IAA and allowed the stress response to develop for two hours. We then subjected the worms to a heat shock (37 °C) for a time period that kills 30–70% of the worms' population (Fig. 5A). Survival chances of trained P0 animals were significantly higher than those of their mock-trained counterparts (Fig. 5B and Supplementary Fig. 4), in line with previous results[3]. F1 progeny of the trained P0 animals also showed significantly higher survival rates when compared to the survival rates of descendants of the mock-trained P0 animals (Fig. 5C and Supplementary Fig. 4). Notably, animals undergoing starvation alone endow their progeny with enhanced survival rates when facing a heat stress[12,62]. In our experiments, both the trained and the mock-trained P0 animals underwent the same starvation regimes, indicating that the inherited acquired changes underlying the odor-induced stress response provide an additional protective layer that enhances survival chances, on top of that already provided by starvation alone (Fig. 5C).

Importantly, the CS IAA alone did not enhance survival chances, as animals which were exposed to IAA, but never experienced starvation, did not show higher survival rates when compared to animals that were never exposed to IAA (Supplementary Fig. 5). Taken together, the mere exposure of the trained animals and their progeny to the CS allowed them to initiate stress response programs in advance, presumably due to odor-induced DAF-16/FOXO nuclear translocation, thus enabling them to better cope with imminent adversities.

### H3K9/H3K36-me, small RNAs, and neuropeptides are involved in the inheritance of cellular changes
As odor-induced avoidance and odor-induced cellular changes (manifested by DAF-16/FOXO translocation to cells' nuclei) are presumably

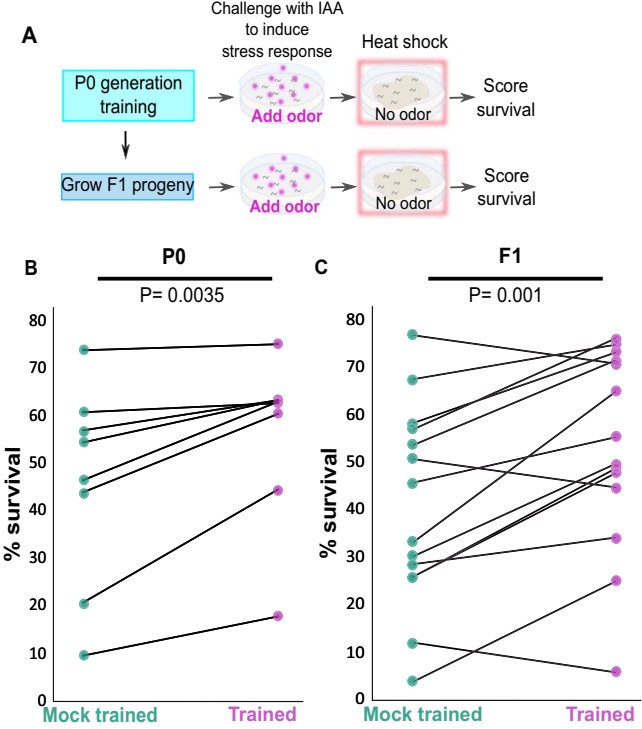

**Fig. 5 | Exposure to the CS IAA increased survival chances upon subsequent stress introduction in both the trained P0 animals and in their F1 descendants.**
**A** The experimental design to quantify survival rates. P0-generation worms were trained (or mock-trained) to associate IAA with starvation. Their F1 progeny were grown under normal conditions. One-day adult P0 and F1 worms were exposed to the CS IAA to initiate the stress response. Following two hours of IAA exposure, the worms were subjected to a heat stress (37 °C for 3.5–5.5 h) and scored for survival the following day (see Methods for more details). Survival rates of trained P0 animals (**B**) and their F1 descendants (**C**) were significantly higher than their mock-trained counterparts after an IAA challenge followed by a heat stress. Note that only the P0-generation animals were trained. Each connected pair of points represents one experimental repeat in which ~50 trained and ~50 mock-trained animals were scored. For P0 (**B**), $N = 8$, p = 0.0035 (one-sided *t*-test, after Bonferroni correction). For F1 (**C**), $N = 14$, p = 0.001 (one-sided *t*-test, after Bonferroni correction). Source data is provided as a Source Data file.

two uncoupled processes (Fig. 4D–F), we continued studying the inheritance mechanisms that mediate the intriguing phenomenon of odor-induced DAF-16/FOXO cellular changes. For this, we analyzed the capacity of animals, defective in key epigenetic genes and pathways, to transmit these acquired cellular changes.

Trained P0 mutants, defective in H3K36 methylation (*met-1*), or in the H3K9 mono/di-methylation (*met-2*) or tri-methylation (*set-25*), showed intact capacities to respond to the conditioned stimulus IAA, but failed to transmit this information to their F1 progeny (Fig. 6A). In contrast, WDR-5.1 and HPL-2 do not seem to play a role in this inheritance, as response levels in both the trained P0 mutants and their F1 descendants were comparable (although lower compared to WT levels, Fig. 6A).

Analyses of the small RNA pathway genes showed that *nrde-3* may be involved in the inheritance of these cellular changes as trained P0 *nrde-3* mutants exhibited WT level responses while their progeny exhibited impaired lowered responses (Fig. 6B). Mutants, defective in the *nrde-2* gene, showed reduced levels of odor-induced DAF-16/FOXO nuclear translocation when compared to WT animals, however, the reduced induction capacity was similar in both the P0 and the F1 animals, suggesting that *nrde-2* may be involved in mediating the acquisition of the cellular changes rather than in their inheritance (Fig. 6B).

*hrde-1* mutants did not exhibit cellular changes in the trained parental generation, thus precluding from assessing the involvement of this gene in the inheritance.

As neuropeptides act cell non-autonomously upstream of the Insulin/IGF signaling pathway to regulate DAF-16-dependent germline proliferation[63], we also studied inheritance of the acquired cellular changes in *unc-31* and *egl-3* mutants that are defective in neuropeptide secretion. While the parental trained generation showed intact responses, resembling those observed in WT animals, these responses were significantly impaired in the F1 descendants (Fig. 6B). Together, these results suggest that H3K9 and H3K36 methylation, the Argonaute NRDE-3, and neuropeptide signaling may be involved in inheritance of the acquired cellular changes.

## Sperm mediates inheritance of the acquired cellular changes

The acquired odor-induced cellular changes could be inherited via the sperm or the oocytes (or both). In all of the above experiments, we trained P0 hermaphrodites which carry both sperm and oocytes, and the analyzed progeny were the outcome of self-mating, thus precluding inference regarding the germline source of these inherited cellular changes. To elucidate which gametes carry the information, we trained either males or hermaphrodites, crossed them with naive partners, and assayed the resulting F1 progeny for the odor-induced DAF-16/FOXO nuclear translocation.

Progeny of naive hermaphrodites that were crossed with trained males showed the typical odor-induced cellular responses (Fig. 7). In contrast, progeny of trained hermaphrodites, crossed with naive males, failed to show these odor-induced stress responses. Importantly, F1 descendants of a cross between trained males and trained hermaphrodites retained the ability to respond to IAA, indicating that mating events do not impair the inheritance (Fig. 7). These findings indicate that acquired cellular changes are transmitted via the sperm.

## The AWC$^{OFF}$ neuron stores the information about the cellular changes in parents and in their progeny

Next, we sought to reveal the neurons that may be involved in storing the information for the odor-induced cellular changes. The CS used herein is IAA and it is sensed primarily by the bilateral AWC$^{ON}$ and AWC$^{OFF}$ chemosensory neurons[64,65], which also participate in coding and storing the information in the parental trained generation[3]. We therefore asked whether these neurons also participate in storing the information in the F1 generation. To address this question, we used two transgenic lines, each exclusively expressing the light-activated channelrhodopsin in either the AWC$^{ON}$ or the AWC$^{OFF}$ neurons. The mere light activation of the AWC$^{OFF}$ neuron, either in the trained P0 animals or in their F1 descendants, sufficed to induce the stress response, as indicated by the rapid translocation of DAF-16/FOXO to cells' nuclei (Fig. 8A). In contrast, light activation of the AWC$^{ON}$ neuron induced DAF-16/FOXO translocation in the trained P0 generation, but not in their F1 descendants. Furthermore, inhibiting AWC$^{OFF}$ activity using the inducible histamine-gated chloride channel during the IAA challenge, abrogated the response in both the trained P0 animals and in their F1 offspring (Fig. 8A). Importantly, histamine by itself did not affect the responses in the P0 nor in the F1 progeny animals (Supplementary Fig. 6). These results suggest that the inherited information is encoded within the AWC$^{OFF}$ neuron of the progeny and that its activity is necessary for the odor-induced stress response.

The possibility to induce stress response by light-activating specific neurons suggests that the information is cell specific rather than stimulus specific. In this case, other stimuli, sensed by the AWC neurons (*e.g.*, benzaldehyde), may be equally used for challenging the worms following training with IAA as the CS. Indeed, using benzaldehyde, the acquired cellular changes could be successfully induced in animals trained to associate IAA with starvation, in both the trained P0 generation and in their F1 progeny (Fig. 8B).

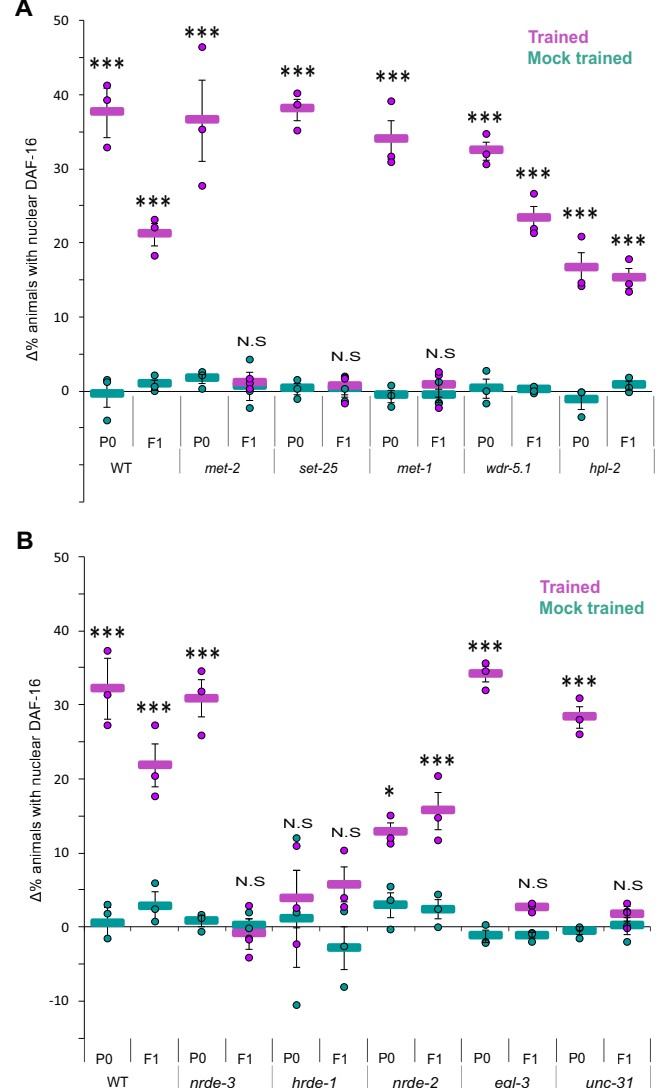

**Fig. 6 | Histone modifications, the small RNAs pathway and neuropeptide secretion are involved in the inheritance of the acquired cellular changes.** **A** Quantification of the acquired cellular changes in histone modifying mutants as evident by the translocation of DAF-16/FOXO to cells' nuclei following exposure to the CS IAA. **B** Quantification of the acquired cellular changes in mutants, defective in genes involved in small RNA processing and neuropeptide secretion, as evident by the translocation of DAF-16/FOXO to cells' nuclei following exposure to the conditioned stimulus IAA. For each group, $N = 3$ biologically independent experimental repeats, each scoring ~50 animals. Significance comparisons are between trained and mock-trained animals. *P*-values from left to right: **A** 6.7E−14, 0.86, 3.8E−19, 0.91, 1.6E−13, 0.42, 7.2E−15, 1.7E−10, 1.8E−6, 1.4E−4. **B** 1.2E−11, 0.62, 0.53, 0.08, 7.8E−3, 5.9E−6, 1.2E−9, 0.17, 5.2E−10, 0.6. *$p < 0.05$, ***$p < 0.0005$ (two-sided proportion test after Bonferroni correction). N.S not significant. Error bars denote SEM. Source data is provided as a Source Data file.

### Serotonin mediates the systemic stress response in parents and in their progeny

In trained animals, serotonin acts downstream to the AWC neurons to mediate the rapid systemic stress response following IAA challenge[3]. We therefore asked whether serotonin also mediates the systemic stress response in the F1 progeny that inherited the cellular changes. Applying external serotonin induced a rapid stress response in both the trained P0 animals as well as in their F1 offspring (Fig. 8C). Moreover, light activation of the serotonergic neurons also induced a rapid stress response in both the trained P0 animals and in their F1 offspring (Fig. 8C). Taken together, the inherited cellular changes are neuron

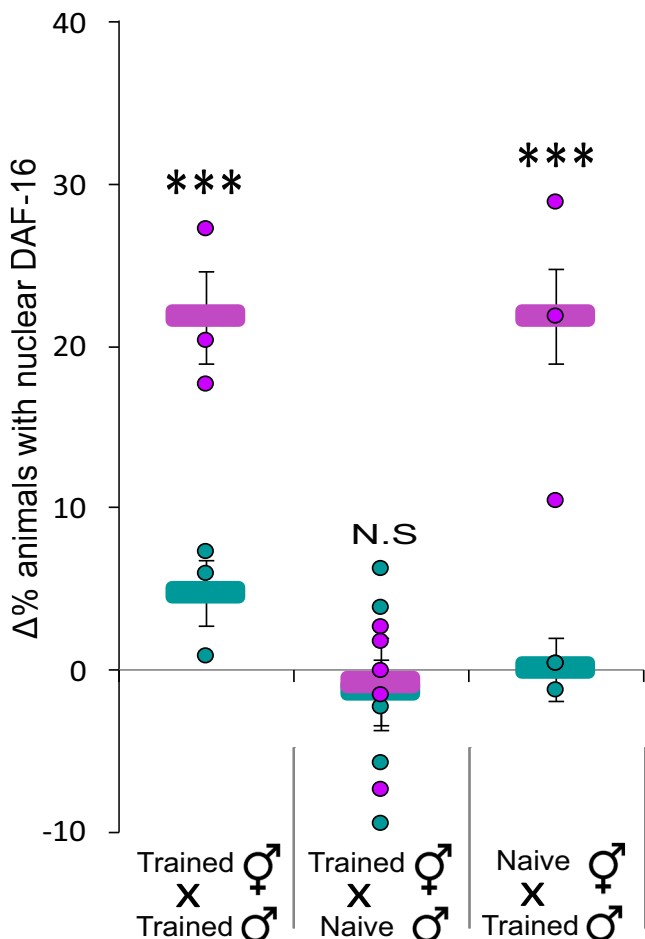

**Fig. 7 | The cellular changes are transmitted to the F1 descendants via the sperm and not the oocytes.** Odor-induced translocation of DAF-16/FOXO to cells' nuclei was observed in F1s which were the progeny of either: (i) trained hermaphrodites crossed with trained males, or (ii) trained males crossed with naive hermaphrodites. F1s progeny of trained hermaphrodites crossed with naive males lacked the odor-induced cellular stress responses. Shown are the means ± SEM of $N = 3$–5 biologically independent experimental repeats, each scoring ~50 animals. *P*-values from left to right: 4.7e−5, 0.14, 3.4e−8. ***$p < 0.0001$ (two-sided proportion test, after Bonferroni correction). Significance comparisons are between trained and mock-trained animals. Source data is provided as a Source Data file.

specific, and the same sensory neuron, AWC^OFF, that stored the information in the trained parental generation also stored the information in the progeny. Furthermore, once the worms are challenged, serotonin signaling mediates the systemic stress response in both the trained parental generation and in their offspring.

## Discussion

We showed that associating a stressful event (*e.g.*, starvation) with an odorant forms an adaptive change that can be transmitted to the progeny. This inheritable associative information is manifested in two outputs: The first is evident by the avoidance of trained animals once challenged with the conditioned odorant. This odor-induced behavioral response may be seen as analogous to the classical memory-evoked responses following fear conditioning typically studied in other vertebrate and non-vertebrate animal models[66,67]. This type of associative memory was transmitted to the F1 progeny, but not beyond. The second output is evident by acquired cellular changes in which the transcription factor DAF-16/FOXO is rapidly translocated to cells' nuclei following exposure to the conditioned odorant. Interestingly, this acquired cellular change was transmitted up to the F2 generation.

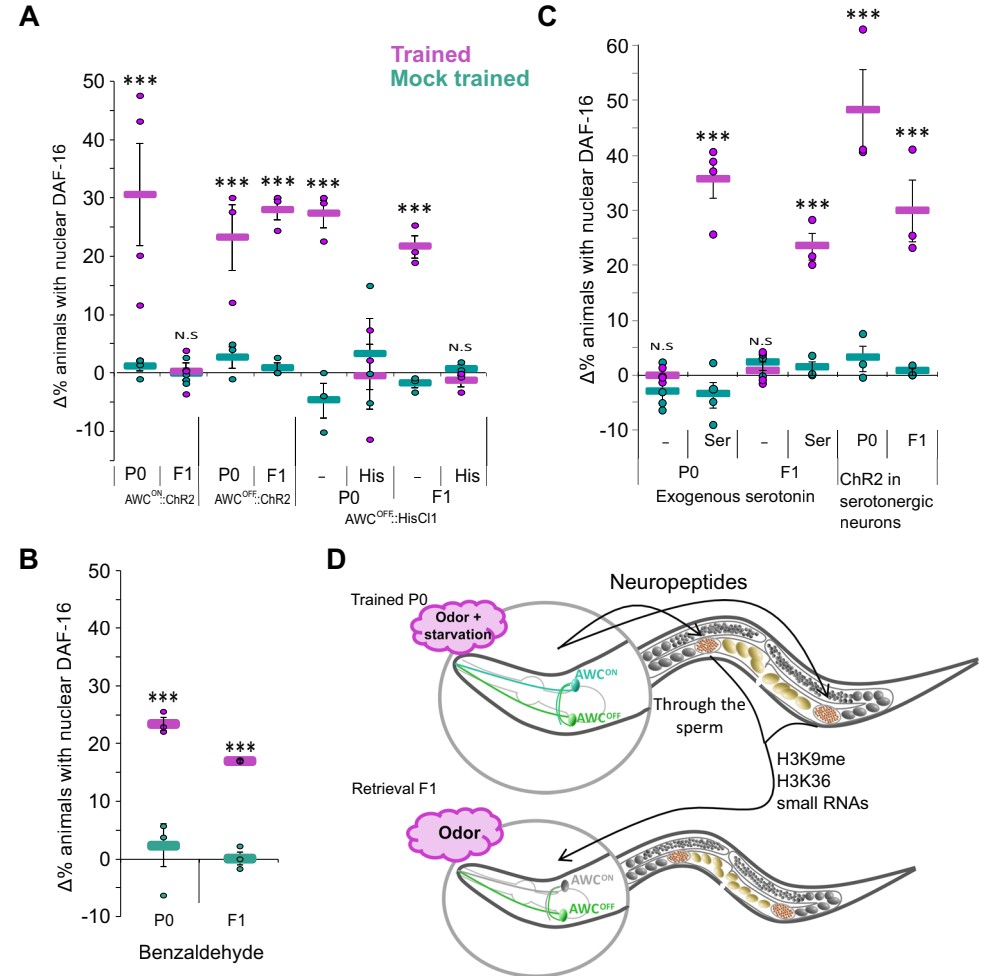

**Fig. 8 | The AWC^OFF neuron stores the information in the parental trained generation and in their F1 descendants. In both, serotonin mediates the systemic stress response. A** Light activation of either the AWC^OFF or the AWC^ON neurons induced the stress response in the trained P0 generation. However, in the F1 generation, light activation of the AWC^OFF neuron, but not of the AWC^ON, induced the stress response. Exclusive inhibition of the AWC^OFF neuron during IAA exposure abolished induction of the stress response in both the parental trained generation and in their F1 progeny. His, histamine added 30 min before challenging the animals with the CS IAA. **B** Benzaldehyde, also sensed by the AWC neurons, can be used to induce the stress response in P0 animals as well as in their F1 progeny. The P0 animals were trained to associate starvation with IAA. **C** Serotonin mediates the systemic stress response in trained animals and in their F1 progeny following odor-induced stress response. Exposure to exogenous serotonin (without odor) induced DAF-16/FOXO nuclear translocation. Similarly, light activation of the serotonergic neurons (ADF, NSM and HSN) induced a rapid systemic stress response in both the trained P0 worms and in their F1 descendants. In all panels, shown are the means ± SEM of $N = 3-4$ biologically independent experimental repeats, each scoring ~50 animals. Significance comparisons are between trained and mock-trained animals. ***$p < 0.0005$. *P*-values from left to right: **A** 1.8E−15, 0.8, 3.3E−5, 1.6E−6, 7.8E−11, 0.7, 1.1E−8, 0.5 **B** 6.8E−8, 4.8E−6 **C** 0.34, 2.3E−26, 0.54, 3.4E−8, 2.8E−24, 9.2E−15 (two-sided proportion test, after Bonferroni correction). **D** A proposed model for inheritance of the acquired cellular changes. In the trained P0 generation, the stress response may be induced by the exclusive activation of either the AWC^ON or AWC^OFF chemosensory neurons, suggesting that these neurons are part of the information-storing ensemble. Neuropeptides, small RNAs, and histone methylation (H3K9 and H3K36) work in concert to modulate the sperm state, and by this, transmit the acquired cellular changes to the somatic cells of the next generation. Notably, while these genes and pathways play a role in the inheritance process, these epigenetic factors are not required for acquiring and activating the cellular changes in the parental trained generation. Source data is provided as a Source Data file.

The behavioral and the acquired cellular changes are presumably two uncoupled processes since odor-induced DAF-16/FOXO nuclear translocation was evident in trained worms which did not avoid the odorant, and worms that avoided the odorant did not exhibit odor-induced DAF-16/FOXO nuclear translocation (Fig. 3D–F). Further support to the notion that these two inherited processes are uncoupled comes from the finding that the memory-induced avoidance was limited to the F1 generation (Fig. 1B) while of the odor-induced cellular changes were evident in the F2 generation (Fig. 3C). Nevertheless, mutant analyses indicated that the same genes and pathways are involved in both inheritance processes (Figs. 2 and 6). Both odor-induced outputs provide the trained animals, and their progeny, with protective means: to physically avoid stressful environments, and in parallel, to prepare in advance for possible dire conditions by quickly initiating stress resistance processes. The latter odor-induced cellular response enhanced animal's survival chances following encountering a subsequent stress, presumably due to odor-induced DAF-16 nuclear translocation (Fig. 5).

For both types of inheritance, we identified the Argonaute NRDE-3 as well as H3K9/H3K36 methylation as potential candidates for the transmission of information between parents and offspring. As small RNAs operate in concert with chromatin modifiers[26,28,32,53], it is possible that memory-specific endo-small RNAs direct the corresponding methyl transferases to establish loci-specific histone marks. In particular, NRDE-3 expression in developing embryos is essential for intergenerational inheritance[56], so it is plausible that RNAs are transferred to (or produced in) the sperm to establish epigenetic modifications in the developing embryos[68,69]. These genes, among others,

had been shown to participate in transgenerational processes to mediate inheritance of stress conditions, such as heat, starvation, exposure to pathogenic agents and others (reviewed in refs. 18,26,30,32).

While memory formation in the parental trained generation did not depend on intact neuropeptide secretion, the inheritance of the memory-evoked avoidance and the odor-induced cellular responses did (Figs. 2B and 6B). Neuropeptides, and particularly secretion of insulin-like peptides, often reflect the physiological state of the animals[70], and were shown to mediate the nutritional status to the germline[12,58,63]. An intriguing possibility is that the two modes of inheritance are established by integrating two seemingly unrelated cues: starvation that is signaled via selective neuropeptides, and the olfactory cue that is encoded by specific small RNAs targeting specific chromatin regions. Indeed, both neuropeptides and small RNAs were shown to be transferred from somatic tissues to the germline to affect the progeny[23,33–35,57]. Furthermore, olfactory adaptation in the AWC neurons is mediated by NRDE-3[71], and stress-induced serotonin signaling modifies germ cells' chromatin to promote their survival[72]. Nevertheless, our experiments using the various mutant strains (histone modifiers, small RNA-related genes and neuro-transmitter/peptide secretion factors) provide merely correlative, rather than causative, indication for their possible involvement in both kinds of inheritances.

As memory-induced behavioral outputs are widely studied, we focused on the less studied process leading to acquired cellular changes and thereof transmittance to subsequent generations. Analysis of mutant strains together with functional interrogation of individual target neurons suggested a possible route for this inheritance (Fig. 8D): The trained P0-generation animals acquire the cellular changes, and the information to trigger the response is stored in both AWC neurons (Fig. 8A and ref. 3). This cellular information is passed on to the next generation where the capacity to induce the stress response is retained in the AWC$^{OFF}$ neuron only (Fig. 8A), indicating that the information-storing neurons only partially overlap between generations. In fact, it is plausible that the information is stored only in the set of neurons required for its retrieval and is excluded from neurons that merely underwent experience-dependent changes in the trained parental generation. In both the trained P0 animals and in their progeny, serotonin mediates the systemic odor-induced stress response.

The acquired cellular changes were robustly observed in F1 animals that hatched from eggs laid 24 h, and even 48 h, post the last training step (Fig. 3C). As a fertilized egg is typically laid within three hours post fertilization[73], the analyzed F1 animals were likely to be in their germline pre-fertilized state during training. Though the germline starts to develop already at the second larval stage, a single training cycle during either the L1, L3, or the L4 larval stages did not lead to inheritance of the cellular changes (Supplementary Fig. 3). Spermatogenesis initiates during the L4 larval stage and is accompanied by substantial transcriptional changes, including CSR-1-mediated licensing and HRDE-1-assisted piRNA-triggered silencing of hundreds of genes, that are required for sperm differentiation and proper function[74]. Yet, while a single training at the L4 stage induced the cellular changes at the parental generation, this information failed to be transmitted to the progeny, indicating that repeated training cycles are crucial for a potent information transmission.

Once the acquired cellular changes were established, they were rather robust as several rounds of exposure to the conditioned odorant in the presence of food were required before these acquired changes went extinct. Even after the acquired cellular changes (as well as the odor-induced avoidance behavior) went extinct in the parental generation, the information could still be successfully transmitted to the F1 progeny (Fig. 4).

The acquired cellular changes were also transmitted to the F2 generation (Fig. 2A). F2 animals did not exist during the training period of the P0 generation, not even in their germ-cell state. They experienced the odorant IAA only once during their pre-fertilized state when their F1 parents were challenged with IAA (Figs. 1A and 3A). This challenge was performed on well-fed F1 animals who never experienced starvation, thus precluding the possibility of coupling between IAA and starvation. Thus, the transfer of the acquired cellular changes to the F2 generation suggests a transgenerational, rather than intergenerational, inheritance mechanism that does not depend on the direct exposure of the germline to the adverse conditions[14,17,18]. An alternative intriguing possibility is that transmittance of the cellular changes depends on challenging and stressing the parental generation. Indeed, we detected odor-induced stress responses only in F2 animals that were pre-selected from F1 animals that showed such odor-induced responses, though F3-generation animals did not inherit the acquired cellular changes, even if they were pre-selected from positive F2 parents (Fig. 3C and Supplementary Fig. 1). In that respect, it may be interesting if inheritance of acquired cellular changes, like other transgenerational processes, is actively regulated to be limited to a few generations only[75,76].

Sperm, but not oocytes, transmitted the epigenetic information (Fig. 7). In mammals, epigenetic inheritance was shown to be mediated by sperm, but the strong bias to perform experiments in male rodents precludes definite conclusions regarding oocytes involvement in such processes[14]. In *C. elegans*, exclusive inheritance via the sperm may have evolved due to their population dynamics constraints. *C. elegans* worms are androdioecious[77], and under non-stressful conditions, hermaphrodites make >99% of the population while the rest are males. However, upon stress, the relative fraction of males significantly elevates, presumably to increase probability of cross fertilization, and by this, genotypic diversity[78]. Thus, cross mating with males that developed through harsh conditions increases the likelihood that their adverse life history will be transmitted to subsequent generations, thereby endowing them a survival advantage. Moreover, the males' sperm outnumbers and outperforms the hermaphroditic sperm, so cross fertilization becomes advantageous as it ensures a wider distribution of the adverse information among the descendants. Indeed, following mating, the sperm contributes to the 1-cell embryo a significant amount of RNAs, particularly of endogenous siRNAs and piRNAs[68,69,79]. Interestingly, siRNAs may trans-generationally enhance mating chances, and by this, spread the epigenetic information[80]. In the event of self-fertilization, the epigenetic information can still be transmitted via the hermaphrodites' own sperm (Fig. 2B).

In summary, here we established that associative memories and acquired cellular changes are heritable and that they provide the progeny valuable behavioral and cellular/physiological protective instructions that ultimately increase their survival chances.

## Methods

### Strains and growth conditions

A full list of the strains used in this study is available in Supplementary Table 1. All strains were maintained and grown under standard conditions[81], unless otherwise indicated, as for example during training or odor-evoked memory retrieval.

### Training the P0 generation to associate an odorant with starvation

We synchronized P0-generation worms by bleaching gravid hermaphrodites to extract fertile eggs[82]. The extracted eggs were seeded on NGM plates that were pre-coated with OP50 bacteria. Following ~18 h in 20 °C, we washed the L1 worms off the plates with an M9 buffer and repeated the washing steps three more times to discard any bacterial residues.

To form the association between the conditioned stimulus iso-amyl alcohol (IAA) and starvation, we used a spaced-training paradigm that overall lasted one week (Fig. 1A). First, the washed L1 worms were transferred to bacteria-free NGM plates (starvation plates), either in the presence of the odorant IAA (trained animals), or in its absence (mock-trained animals). The starvation plates contained 2.5–3% agar to minimize worm burrowing, and ampicillin (100 µg/mL) to prevent possible contamination and bacterial growth. The IAA odorant was diluted 1:1000 and was added by applying 9 equally distant 5 µL drops on the inside face of a 50 mm plate's lid (the trained worms plate).

After 24 h, the worms were washed off the plates and transferred to fresh OP50-seeded NGM plates for a period of ~6 h to allow recovery from starvation. A second training was then imposed by washing the worms three times in M9 and transferring them (trained and mock-trained groups) to starvation plates for over-night conditioning in the presence (trained) or the absence (mock trained) of IAA. At this stage, the worms experienced starvation as L2-L3 larvae. Following 18 h of starvation, the worms were washed off the plates and allowed to recover on food (~6 h), after which they were washed again and placed on starvation plates (with or without IAA) for an additional 18 h (third starvation). We then recovered the worms for 4–8 h on food. After this step, the majority of the worms reached the 4th larval stage (L4).

We then subjected these L4 worms to a fourth and final ~65 h long starvation (with or without IAA), followed by recovery on OP50-seeded NGM plates for 24 h. By this time, the worms reached the young adult stage and started to lay eggs. About 60–70 worms, from each of the trained and the mock-trained adult P0 groups, were randomly selected to validate successful training and for further analyses.

### Training worms on food
To control for the possibility that the mere exposure to IAA may induce DAF-16/FOXO nuclear translocation, we trained the worms to associate IAA while on food. Since animals trained on food develop faster than animals trained with intermittent starvations, we used a similar training protocol as shown in Fig. 1A, but adjusted the IAA exposure schedule to the faster growth rate. Specifically, 24 h post hatching, we added IAA for 24 h. Worms were then washed off the plates and incubated for 6 h in the absence of IAA, followed by an additional exposure to IAA for 18 h, a wash, and 6 h of recovery in the absence of IAA. At the end of this last step, the worms had reached the young adulthood stage and were subsequently assayed for DAF-16/FOXO nuclear translocation following IAA challenge (Supplementary Fig. 2A).

### Training specific larval stages
We also trained worms to associate IAA with starvation exclusively in either the L1, L3, or the L4 larval stages. For the L1 stage, worms were starved in the presence of IAA for 24 h immediately post hatching. For the L3 stages, we started starving the worms in the presence of IAA 24 h after bleaching (most animals reached the L3 stage by then). This starvation lasted for 24 h. For the L4 stage, we starved the worms in the presence of IAA for 24 h, 48 h post bleaching. After the 24 h of starvation, the worms were transferred to fresh OP50-seeded NGM plates and allowed to reached adulthood. All the F1 progeny of the trained P0 animals were grown on food and never experienced starvation nor were they exposed to IAA.

### Retrieval of the F1 generation
We allowed the trained/mock-trained P0 generation to lay eggs for 24 h during the last recovery period, after which we washed the plates with M9 to remove the P0 worms as well as F1 descendant larvae that already hatched. Following the washing step, we verified that only F1 eggs remained on the plate. Notably, washing off larvae that hatched during the 24 h after the recovery period ensured that the F1 progeny remaining on the plate for analysis were still in the unfertilized germ state during the training of the P0 worms. After the eggs hatched, the

F1-generation worms were collected and transferred to fresh OP50-seeded NGM plates for further growth. Once reaching adulthood, about 60–70 F1-generation worms from both trained- and mock-trained P0 worms were randomly selected for analysis.

### Retrieval of the F2 and the F3 generations
For F2-generation analysis, we allowed the F1-generation worms to lay eggs, and then washed the worms off the plate, leaving only unhatched eggs on the plate. After the eggs hatched, these F2-generation worms were transferred to fresh OP50-seeded NGM plates for further growth. Upon reaching adulthood, we randomly picked 60–70 of the F2 worms (descendants of each trained and mock-trained P0 worms) for further analysis. Similarly, we picked and assayed the F3 generation.

### Avoidance assays
For behavioral assays, we used WT N2 worms or the various mutant strains (not expressing the Pdaf-16::DAF-16::GFP transgene). We trained the P0-generation worms to form an associative aversive memory as described above (also Fig. 1A). Individual trained or mock-trained animals (either the P0 or the F1 generations) were transferred to unseeded plates and allowed to adjust for ~2 min before the assay. An individual worm was tracked by eye, and when inspected to perform a forward locomotion for few seconds, a thin stripe of the odorant IAA (1:1000 dilution in water) was applied in front of the worm head, perpendicular to its forward trajectory. When applied, we verified that the IAA did not come in contact with the worm. Worms that stopped in front of the IAA and backed within three seconds were scored as avoiding; worms that only briefly stopped, or continued crawling forward to cross the IAA stripe were scored as non-avoiding.

To test F1 animals, pre-selected from avoiding P0-generation animals, we transferred ~60 P0-generation worms from each of the trained and mock-trained groups that were scored as avoiding to fresh OP50-seeded NGM plates, and allowed the worms to lay eggs for 24 h. We then washed the plates with M9 to remove the P0 worms as well as the F1 descendant larvae that already hatched. Following the washing step, we verified that only F1 eggs remained on the plate. These F1-generation worms were assayed as described for the P0-generation worms.

### Mating procedures of trained and naive P0 worms
To determine whether the inheritance is paternal or maternal, we used the ZAS394 strain, which, along with the DAF-16 translational reporter (daf-16::DAF-16::GFP[83]), also expresses the pan-neuronal red fluorescent marker (rab-3::tagRFP-NLS[84]) in the background of an him-5(e1490) mutant to increase male counts in the population. We trained either TJ356 (for maternal inheritance) or ZAS394 (for paternal inheritance), or both, and after 24 h of recovery period, we transferred 50 ZAS394 males (either naive or trained), and 15 TJ356 hermaphrodites (either trained or naive, respectively), to a large mating plate (at least 5 large mating plates per experiment). We allowed the worms to mate, and after three days we picked rab-3::tagRFP-NLS expressing offspring, to verify that they are the result of a successful mate. Those descendants were analyzed upon reaching adulthood as the F1 generation.

### Quantifying odor-induced stress responses based on DAF-16/FOXO translocation to cells' nuclei
Here, we followed the detailed procedures provided in[3]. To quantify the induction of the stress response, we used worms expressing a translational fusion of the general stress response transcription factor FOXO/DAF-16 (pdaf–16::DAF-16::GFP[83]). A worm was scored as stressed if we clearly identified at least six cells with DAF-16::GFP nuclear localization. The cutoff of six cells was carefully selected based on our previous extensive analyses showing that this is a conservative estimation of the stressed worms in the population[3]. We focused on

scoring DAF-16::GFP nuclear localization in the gonad sheath cells as this tissue was the first and the most prominent to show this spatial dynamic pattern.

Each experiment always included two groups: trained and mock-trained animals. We first quantified the baseline stress levels in both groups by scoring the % animals that show nuclear DAF-16/FOXO before the challenge. These values were then subtracted from the stress levels quantified in the same groups following the IAA challenge. This subtraction was conducted for each experimental repeat independently and these results were then averaged. When worms did not initiate stress responses following exposure to the CS IAA, the % of stressed worms was very close to (and sometimes even lower) than the % observed in the pre-challenged worms. Therefore, negative values were sometimes obtained when calculating the % stress in mock-trained animals, or when assaying conditions/mutants that did not elicit stress in trained animals.

Importantly, following the last recovery step from starvation, and prior to challenging the worms with IAA, we verified that worms had recovered from the starvation-induced stress. This was done by visual inspection of the DAF-16::GFP protein, ensuring that its spatial localization within the cells was cytoplasmic (rather than nuclear).

All scoring procedures were done in a blind manner such that the examiner did not know the treatment that the worms underwent (e.g., trained or mock trained). Each scoring plate consisted of at least 50 animals that were inspected under a fluorescent binocular (MVX10, Olympus) with a high-zoom magnification (X300). Typically, each data point is the average of at least three independent experimental repeats, each performed on a different day.

For stress induction using light-activated ChR2, we recovered the worms from starvation on plates pre-seeded with OP50 containing All-Trans Retinal (100 μg/ml, Sigma). Following the recovery period, we exposed the worms (while on the same recovery plates) to 488 nm blue light for 20 min before scoring. The light source was the LED coupled to the fluorescent binocular. Notably, the blue light on its own did not induce stress, as no nuclear DAF-16::GFP was observed in the mock-trained animals, nor in their F1 and F2 progeny.

To inhibit neural activity via the histamine-gated chloride channel[85], we placed the worms on plates pre-seeded with OP50 supplemented with 10 mM histamine (Sigma). The worms were allowed to absorb the histamine for at least 30 min before challenging with IAA.

In serotonin feeding experiments, the worms were placed on plates pre-seeded with OP50 mixed with 10 mM serotonin (Sigma). The worms were first inspected to verify that the mere handling and transfer did not cause DAF-16 to translocate to cells' nuclei. The animals were then allowed to absorb the serotonin for 20 min, after which we scored for DAF-16 nuclear localization.

## Heat-shock induced survival assays
We trained WT N2 hermaphroditic worms (as detailed above) which constituted the P0 generation. To induce the odor-evoked stress response prior to the heat shock, P0 or F1 worms were transferred to fresh plates without food and presented with IAA ($10^{-3}$) for two hours by applying 9 equally distant 5 μL drops on the inside face of the plate cover. The worms were then transferred to OP50-containing plates with no IAA and subjected to a 37 °C heat shock, a stress that was partially lethal to animals. As the inflicted % death considerably varied across experimental days, we analyzed varying durations of heat shocks. Notably, we always compared survival rates between same-day trained and mock-trained animals, where both groups were incubated together for the exact same duration at 37 °C. The P0-generation worms were heat shocked for 4.5–5.5 h while the F1-generation worms that were never starved were heat shocked for 3.5–4.5 h (Supplementary Fig. 4). The extended heat shock period used for the P0 generation was due to their higher resistance to heat,

presumably due to the intense stress (repeated starvations) that they underwent throughout development. Heat shock and survival assays quantified % survival of trained and mock-trained animals held separately (and not by direct competition). Since heat shock induces quiescence, it is difficult to assess mortality immediately after the heat shock. We therefore scored viability on the following day, when surviving animals were clearly motile, pumping, and withdrawing following a gentle touch on their head. Animals not showing any of these signs were scored as dead.

## Quantifying DAF-16 expression levels
To quantify the expression levels of DAF-16, we quantified the fluorescence of individual animals expressing the P*daf-16*::DAF-16::GFP transgene[83]. We compared expression levels of naive, trained, and mock-trained worms, of both P0 and F1 generations (Supplementary Fig. 2B). Worms were immobilized using levamisole and mounted on a slide covered with a thin agar pad. Imaging was performed using an Olympus IX-83 inverted microscope equipped with a CMOS (Photometrics) camera at a ×10 magnification. Fluorescent intensities of individual animals were extracted using in-house Matlab scripts.

## Statistical analysis
We used the proportion test to infer statistical significance between the different groups and conditions. For this, we considered two pairs of proportions ($p_b^{c1}, p_a^{c1}, p_b^{c2}, p_a^{c2}$), where each pair represents the proportion of the stressed worms before (*b*) and after (*a*) the challenge in the two compared conditions (*c1, c2*; e.g., trained and mock trained). This analysis tests whether the difference between the first condition proportions ($p_a^{c1} - p_b^{c1}$) is likely to be sampled from the expected distribution of the difference between the second condition proportions ($p_a^{c2} - p_b^{c2}$). Formally, the null hypothesis is defined as:

$$(p_a^{c1} - p_b^{c1}) \sim N\left(p_a^{c2} - p_b^{c2}, \sqrt{\frac{p_a^{c2} \cdot (1 - p_a^{c2})}{n_a^{c2}} - \frac{p_b^{c2} \cdot (1 - p_b^{c2})}{n_b^{c2}}}\right)$$

Where *n* is the total number of analyzed animals in this condition.

## Reporting summary
Further information on research design is available in the Nature Portfolio Reporting Summary linked to this article.

## Data availability
All the data is supplied in the manuscript. Source data are provided with this paper.

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

## Acknowledgements

We thank Itamar Harel and Yonatan Tzur for insightful comments on the manuscript draft. We are also thankful to Ehud Cohen, Meital Oren, the Mitani lab, and the CGC for strains. The CGC is funded by the NIH Office of Research Infrastructure Programs (P40 OD010440). This work was supported by ERCStG (336803), ICORE (1200/12), and ISF (1300/17). N.D. is a Clore Scholar and is grateful to the Clore Israel Foundation. A.Z. is the Greenfield Chair in Neurobiology.

## Author contributions

N.D., Y.E., L.H., and S.B.E. performed the experiments. E.I. and E.B. developed statistical and analytical methods. N.D., Y.E., and A.Z. analyzed the data and wrote the paper.

## Competing interests

The authors declare no competing interests.
