## [Peer Review File · Nature Communications]

Inheritance of associative memories and acquired cellular changes in *C. elegans*REVIEWER COMMENTS

Reviewer #1 (Remarks to the Author):

In this paper, Deshe and colleagues describe the inheritance of associative memory across two generations and elucidate part of the underlying molecular and cellular mechanisms in *C. elegans*. Using an established associative learning paradigm, they first show that associative memory of odor-starvation pairing during larval development is transmitted to the progeny for up to two generations. This inherited memory is restricted to a physiological conditioned stress response, assessed by the nuclear translocation of the DAF-16/FOXO transcription factor, whereas a behavioral conditioned response is not inherited. The authors show that memory reactivation, when presenting the conditioned stimulus (the odor isoamylalcohol) to the parental or F1 generation, protects the animals from heat stress. By testing mutants defective for methylation, small RNA pathways and neuropeptide signaling, they identify a number of mutants that are specifically defective in memory inheritance. The authors then investigate whether the cellular basis for associative memory as characterized previously in the parental animals (Eliezer et al., 2019) is similar to that of memory in the F1 generation. They show that the odor-sensing neuron AWC OFF and downstream serotonin signaling is indeed required for memory retrieval in both the P0 and F1 generations.

The authors have presented a clear story and the identification of an inherited associative memory is a valuable contribution to the study of learning and memory. The manuscript clearly describes the experimental details and is overall well-written. However, the mechanism of memory inheritance remains unclear and several concerns need to be addressed to substantiate the authors' conclusions.

Major comments:

1. The authors state that “the inherited memory endowed the progeny with a fitness advantage”. However, it is unclear whether the experiment shown in Figure 2 was a competition experiment between trained and control animals, which would assess a potential fitness advantage, or whether heat stress resistance was quantified in trained and mock-trained animals separately, which would not directly assess fitness. The authors should clarify this and adjust their statements accordingly. Additionally, it is not clear why there is a large variation and much higher N number for the experiments with the F1 animals.
2. Several mutants were identified that are specifically defective in memory inheritance, leading the authors to state that H3K9/H3K36 methylations, NRDE-3 and neuropeptide secretion are required for the inheritance. However, only one mutant allele of these genes has been tested; thus, it cannot be excluded that other genetic changes (e.g. background mutations) may be responsible for the observed inheritance defects in these mutants. Additional evidence is needed to show that these genes are indeed required for memory inheritance.
3. The authors identified a number of candidate genes that could be involved in transmitting associative memory to the progeny. Overall however, the mechanism underlying memory inheritance, including the molecular identity of the transmitted signal(s) and the cellular mechanisms, remains unclear. For example, the cellular focus of the identified genes has not been investigated. Are these players, e.g. neuropeptide signaling, required within the memory circuit including AWC and/or serotonergic neurons, or are other cells/tissues involved?
4. Supplementary figure 2 shows that exposure to IAA in the presence of food does not induce DAF-16 translocation. This is an important control in the associative memory paradigm. However, the experimental details on the training protocol and the duration of IAA exposure as compared to the IAA-starvation pairing seem to be missing. This information should be included in the manuscript, and IAA exposure should be comparable to the starvation training protocol. It would also be helpful to describe the results of this experiment in the main text.
5. Odor-evoked memory reactivation triggered both behavioral and physiological responses in the

trained P0 generation, but only the physiological stress response seems to be inherited in the F1 and F2 generations. For this physiological response, F1 animals had to be pre-selected to see an effect of memory reactivation in the F2 generation. Could inter-animal variability also mask inheritance of the behavioral response? Around 45% of P0 animals avoided IAA after training (Fig 1F), but the authors did not test behavioral responses of F1 animals directly descending from these animals showing the learned behavior.

6. The figure legend of supplementary figure 1 states that the overall stress in the population without IAA exposure was assessed by quantifying the % of worms with nuclear DAF-16 before IAA challenge. This value was subtracted from the % of worms with nuclear DAF-16 after the IAA challenge, and the same subtracted values were used in all other figures. Based on this statement, it is unclear whether the overall stress in the population was assessed for each experiment independently, as basal stress levels may vary between different experimental repeats.

7. Figure 4B: Histamine-induced silencing of the AWC-OFF neuron blocks retrieval of the conditioned stress response in P0 and F1 animals. However, the authors did not test whether histamine treatment alone, in the absence of the histamine-gated chloride channel, affects DAF-16 localization. This control is crucial to ensure that the observed effect is due to silencing of AWC activity rather than histamine exposure.

Minor comments:

1. Most experiments, except for Figure 1, were only done for the F1 generation (not F2s). In the methods section, the authors mention that the analyzed F1 progeny were still in the unfertilized germ state during training of the P0 animals. Thus, F1 animals were not exposed to the training conditions in the embryonic stage. This information is important for the correct interpretation of data in Figures 2-4, but it is not clear from the main figures. It would help to clarify this in the experimental scheme of Figure 1A.

2. The authors cite the famous experiment of Pavlov as an example of classical conditioning. The original studies describing these experiments, however, did not mention the use of a bell, even though this is commonly stated as such. It would be better to rephrase this example, f.i. as an association between food and a sound or auditory cue.

3. Lines 46 and 48: *Pseudomonas* should start with a capital letter and should be written in italic.

4. Fig 1B-D lack scale bars.

5. Line 147-148: "Heat shock typically kills around 50% of the worm population." Does this refer to well-fed N2 animals or animals after odor-starvation training? This is not clear from the manuscript, because Fig. 2 shows a survival rate ranging between 10% and 80%.

6. Line 175 and 231-232: naïve should be naive.

7. Line 203: "that were shown to function in..." (in is missing)

8. Lines 206 and 219: state that *Argonaute* and neuropeptide secretion mutants show intact WT levels of memory formation and retrieval. However, Fig. 3B-D do not include WT control animals, so these statements are not supported.

9. Fig. 3C: mutants of *nrde-2* seem to show even stronger stress responses in the F1 generation than the P0 animals. Is this difference statistically significant?

10. Line 259: "Remarkably, the mere light activation of AWC-OFF [...] sufficed to induce a rapid stress response". The authors' previous work already showed that the AWC-OFF neuron is necessary and

sufficient to trigger the stress response, so this finding is not remarkable or unexpected, but in line with previous reports.

11. Lines 317-318: “a combination of [...] and neuropeptide secretion may mediate...”

12. Supplementary Table 1: unc-31 mutant is missing from the table.

Reviewer #2 (Remarks to the Author):

This paper builds on an aversive association assay previously published by the same lab (Reference 3 in the reviewed paper). In the previous work they detailed the learning assay. In this paper they show that the associative memory established in the P0 generation (previously published work) can be passed (at low levels) to the F1 and F2 generations. Furthermore, they show that the inheritance requires some histone methyltransferases and neuropeptide secretion. They went on to pinpoint differing requirements for AWCOFF and AWCON neurons in the signaling pathway. Given that these neurons had already been implicated in the pathway in their previous paper, these findings aren't quite as novel as they would otherwise be.

The work will be of significance to the field, and most conclusions drawn are sound, but there are additional experiments that the authors could have performed that would make the paper more thorough.

The only conclusions that I believe are wrong are those relating to the roles of the various genes tested in the assay (Figure 3B-D). In Figure 3C line 241/2 “mutants defective in the Argonaute hrde-1 showed impaired memory... thus precluding inference of their role in the inheritance”. This is true, but is surely also true to a lesser degree for nrde-2 and hpl-2. Please adjust your conclusions.

Line 198-201 Is the wdr-5.1 F1 level significantly lower than WT? IF so, I think that argues that WDR-5/1 does play a role in memory inheritance, albeit not as strong a role as met-2 and set-25. Similarly, (Lines 207-211) nrde-2 is here described as having intact capabilities for memory retrieval in P0 and F1 but appears to have substantially lower memory in P0 compared to WT (not shown on the graph). Figure 3D again, egl-1 mutants do look slightly defective at P0. Comparisons with wildtype should be performed.

Additional experiments that would improve the paper.

Figure 1E and Supplementary figure 3 deal with the inheritance of the associative memories. In figure 1E the authors mix data from selected parents (F2) with that from unselected parents (F1). They then in Supp Fig 3 infer what they think the inheritance rate is from P0-F1 without actually doing the experiment. It would be better if they actually selected P0 animals that display the learning to produce the F1 generation. Then they can accurately quantify inheritance properly. Doing this would also allow them to sort out the role of hrde-1 and hpl-2 in learning vs memory

Also in Figure 1E I was confused as to how the P0 untrained animals exhibited a negative percentage of nuclear DAF-16, and what is the delta on the y axis referred to until I read the Figure legend in Supp figure 1. I recommend including some of that info in the figure legend for Figure 1.

Figure 3A and Supplemental figure 4. The finding that training only at the L4 stage does not allow for memory transmission to progeny is striking, but a bit unsatisfying. It would have been nice to see this extended – the training consists of 4 periods of starvation that occur throughout larval development. It would be illuminating to train populations for each of these training steps individually in order to determine which of the larval stages are important for establishing memory (especially since this is talked about in the discussion). It is also important to note that only the offspring laid (but not hatched) during the first 24 hrs of P0 adulthood were assayed. It would be interesting to wash the adults to fresh plates and then collect the second 24 hr batch of offspring. Would the pattern of memory be different? Would memory then be observed in the offspring of L4-only trained animals?

Other issues

Line 58 I could be wrong, but I thought the number of neurons is now officially down to 300?

Starvation training – how many of the worms survived this starvation procedure? That's quite a high level of starvation, was there a severe population bottleneck that occurred? Similarly, how dramatically was the fertility affected by the starvation training? Could this have implications for memory and learning?

Figure 1 B/C/D the images supplied are low resolution and low magnification, making it very difficult to see the nuclei that we are supposed to be able to see. Higher magnification inserts with better resolution should be supplied. Can the nuclei be identified in some other way also? DAPI or showing the brightfield/DIC image? As another note- the worms appear to have a dumpy phenotype, which is particularly evident in D. This is not mentioned. Why are they dumpy?

Figure 2B why is there such a huge range in survival in the P0 generation? Ranging from only 10% surviving to 75% - is it to do with the range of time that the heat shock was performed? 4.5-5.5 hrs for P0 is actually quite a substantial range. If the authors plot length of heat shock vs survival do they see a trend? I appreciate that this isn't the point of the paper, but it seems that there is a broad trend that the difference between trained/mock worms is larger when the mortality rate is more severe and so perhaps the data would be tighter and more convincing with a smaller spread of mortality.

Figure 3, all panels: what do the **** refer to? Which comparison is significant?

Line 259/260 It is unclear whether the P0 and F1 worms were light activated, or if it was only the P0 worms. Wording should be fixed to clarify.

Reviewer #3 (Remarks to the Author):

In the manuscript "Inheritance of associative memories in *C. elegans* nematodes" Deshe N et al developed a paradigm for testing the inheritance of starvation memory in *C. elegans* by associating starvation with the odorant isoamyl alcohol. They demonstrate through some elegant work that this starvation induces a translocation of DAF-16 which can be transmitted for 2 generations. Furthermore this causes a heritable increase in heat stress survival and is mediated by a paternally transmitted signal which requires the putative H3K9 methyltransferases MET-2 and SET-25, the putative H3K36 methyltransferase MET-1, the argonaute NRDE-3, as well as the neuropeptide signaling pathways involving EGL-3 and UNC-31. Finally they perform some experiments using different mechanisms of activating specific AWC neurons to demonstrate that these also have intergenerational effects on DAF-16 localization in the particular cells. All in all I thought this was an excellent and novel manuscript and while more work could be done to demonstrate a deeper mechanistic understanding of this phenomena I'm not sure if that is really required for this initial story. I think that most of the changes that are required are the addition of controls and altering the presentation of the data to more accurately reflect the findings.

Major Points

1) The majority of this manuscript hinges on DAF-16 localization. Since many treatments will cause DAF-16 localization changes the authors should be exceptionally careful about drawing a direct line between each of them. For instance the AWC results could be an independent mechanism from the starvation results. Since this is the only proxy for the phenomena if an additional metric could be added to bolster the findings that would greatly strengthen the findings. Additionally it is important to represent all these plots as dot plots with each individual experiment represented by a different dot on the bar graph so the readers can judge the magnitude and consistency of the phenotype. I would also include numbers in the text so that readers can judge the magnitude of the phenotype in addition to visualizing them.

Minor Points

- 1) *C. elegans* are nematodes so it is redundant to have both in the title
- 2) There are more than 3 mechanisms that could underlie transmittance of epigenetic information.

Beyond DNA methylation, chromatin modifications and small RNAs there are also microbiota, mitochondria or maternally inherited proteins, prions... Those 3 have been the best studied but there is no indication that they are the main three.

3) Fig 1 B-D needs to be labeled on the figure itself what are the conditions and what is being assessed should all be visible in the figure. For 1E (and all of the subsequent plots) it would be best to do a dot plot to represent each of the individual experiments so readers can visualize the magnitude of the differences rather than just representing the mean +/-SEM.

4) As a negative control the authors should show that the odor itself does not alter daf-16 localization.

5) The color in Fig 2B+C is unnecessary and distracting. The authors should represent the heat stress survival data as a traditional bar plot with dots representing each experiment and include statistics. Additionally in this section it would be good to cite the work that has shown that starvation in the parents leads to increased heat stress survival in the descendants (Jobson et al 2015, Webster et al 2018)

6) It is probably worth mentioning that met-2, set-25, and wdr-5 have all been implicated previously in inheritance of non-genetic information in *C. elegans* and including the appropriate citations.

7) It is a good idea to cite the papers demonstrating that unc-31 and egl-3 are defective in neuropeptide secretion pathways when describing that you are testing those genes.

8) In the discussion it is important to state that these conclusions are based on correlative experiments. The fact that genes involved in H3K9, H3K36, small RNA inheritance, and neuropeptides are important for the phenotype does not ensure that those molecules themselves are causal. I think if the authors wish to state causality a lot more work would be required. This is not necessary for this manuscript but a tempering of the conclusions is.

Reviewer #4 (Remarks to the Author):

In this study the authors investigate whether aversive associative memory can be inherited using the *C. elegans* model system. The unconditioned stimulus was starvation and the conditioned stimulus was isoamyl alcohol (IAA) – a normally attractive odorant. The training protocol for P0 animals consisted of inducing 4 periods of starvation at each larval stage (beginning at the L1) in the presence of IAA which lasted anywhere from 12-64 hours. Starvation periods were followed by a 6-24 hour recovery period in which the animals were on food and not exposed to IAA. The authors used translocation of GFP-tagged FOXO/DAF-16 into the nucleus of gonad sheath cells as an indicator of stress response activation. Trained P0 animals showed DAF-16 nuclear translocation after the mere exposure to IAA post training whereas mock-trained control animals did not. The F1 and F2 progeny of the trained P0 worms also showed DAF-16 nuclear translocation when exposed to IAA without ever undergoing the training protocol. In trained P0 animals and their F1 progeny, the induced stress response from IAA exposure increased survival rates following heat shock. The authors then investigate mechanisms mediating the transgenerational inheritance of the aversive associative memory and found that the memory is passed via the sperm. By training and testing animals and their F1 progeny that carrying knockout alleles of genes involved in small RNA pathways and chromatin modification processes, the authors determine that methylation of H3K9 and H3k36 as well as the NRDE-3 gene are important for the inheritance of the associative memory. The authors then determine that genes involved in neuropeptide synthesis and release are also important for memory inheritance. Using optogenetics and a histamine-gated chloride channel, the authors determine that the aversive associative memory is encoded by the AWC neurons, and activation of the AWCoff neuron in the F1 progeny of trained worms could induce DAF-16 nuclear translocation. Optogenetic activation of serotonergic neurons in trained P0 animals and their F1 progeny, or addition of exogenous serotonin, could induce nuclear translocation of DAF-16, suggesting serotonin plays a role in the activation of the aversive associative memory. The authors conclude that

Summary

- What are the major claims of the paper?

This paper claims that, in *C. elegans*, aversive associative memories can be inherited to provide offspring with a fitness advantage through activating the stress response. The authors use mutant

analysis, optogenetics, and other genetic techniques to indicate the inheritance of the physiological response memory is passed via the sperm, relies on neuropeptides, and genes involved in chromatin modification and small RNA pathways – whereas the associative memory is encoded by AWC off neuron and optogenetic activation of this single neuron or serotonergic neurons, or exogenous application of serotonin, can activate the inherited physiological response memory in the progeny.

- Are the claims novel? If not, please identify the major papers that compromise novelty

As the authors state, previous literature has indicated that transgenerational inheritance of different forms in different model systems. Inheritance of other types of memories have been shown to be inherited via sperm (<https://www.biorxiv.org/content/10.1101/2020.11.18.389387v2>). The genes chosen for mutant analysis have well-known roles in key pathways involved in epigenetic inheritance. While the authors were able to conclude which of these genes are important for the inheritance of the physiological response memory, no novel genes or pathways were investigated. Further, many additional lines of inquiry such as which neurons are involved and serotonin's involvement in stress response induction have been explored in the lab's previous publication regarding aversive associative memory (<https://doi.org/10.1016/j.cub.2019.03.059>). Therefore, the main contribution of this paper is confirming whether these genes/mechanisms are also involved in inheriting and/or inducing the physiological response induced by the aversive associative memory tested in the F1 generation.

- Will the paper be of interest to others in the field? Yes

- Will the paper influence thinking in the field? If correct controls are run.

- Are the claims convincing? If not, what further evidence is needed?

Key control experiments are missing and many reasonable lines of inquiry were not addressed.

- Are there other experiments that would strengthen the paper further? How much would they improve it, and how difficult are they likely to be?

See comments below.

- Are the claims appropriately discussed in the context of previous literature?

The authors need to more clearly address how their work relates to their own previous publication and the work of other major *C. elegans* researchers (such as Dr. Oded Rechavi, Dr. Coleen Murphy, and Dr. Catharine Rankin).

- If the manuscript is unacceptable in its present form, does the study seem sufficiently promising that the authors should be encouraged to consider a resubmission in the future?

Yes – however they need to add behavioral data for all mutant lines tested, not just *daf-16* translocations as described below.

Is the manuscript clearly written? If not, how could it be made more accessible?

The manuscript is written very clearly and the experiments are well described in the methods section.

There are only a few minor comments about figure formatting or labeling below.

- Could the manuscript be shortened to aid communication of the most important findings?

No, if anything the authors need to provide more information to support their claims.

- Have the authors done themselves justice without overselling their claims?

The authors are overselling their claims, as there was no inheritance of a behavior, only a physiological response. Further, while the authors were able to show the involvement of genes involved in epigenetic changes, neuropeptide production/secretion and serotonin release were involved in memory re-call/inheritance, the authors provide very little novel evidence of the underlying mechanisms being affected/activated.

- Have they been fair in their treatment of previous literature?

The authors probably could provide more information about the findings of previous literature on inheritance of other forms of memory or physiological responses. Further, it seems strange that work from major *C. elegans* researchers who focus on memory or transgenerational inheritance were not examined in paper's discussion. Are there any major mechanistic differences between the inheritance of associative versus non-associative memories?

Have they provided sufficient methodological detail that the experiments could be reproduced?

The methods section provided sufficient detail to allow others to replicate their procedures.

Is the statistical analysis of the data sound?

Each experiment usually included multiple experimental replicates which scored 50 worms each.

Were stats done on individual replicates or on cumulative data? If cumulative did each replicate give

the same results? The authors chose to use the proportion test for their analysis but do not give a rationale for the choice. From <https://www.itl.nist.gov/div898/handbook/prc/section2/prc24.htm>: Because the test is approximate, N needs to be large for the test to be valid. What is the N used for each analysis (is the N per group or the overall N for all of the animals compared). Does the analysis take into account multiple tests on a single experiment, ie for figure 3B there are 10 groups (5 comparisons) that must have been tested- how is this accounted for?

- Should the authors be asked to provide further data or methodological information to help others replicate their work? (Such data might include source code for modelling studies, detailed protocols or mathematical derivations).

- Are there any special ethical concerns arising from the use of animals or human subjects? No

Major comments

1. Memory is a behavioral construct, not a physiological process (those would be described as cellular mechanisms underlying memory or cellular correlates of memory)- without behavioral data at best this might be a correlate of memory- not the memory itself- the authors lose track of this and consider the daf-16 to be the memory. (It took almost 10 years to convince people that LTP was a real reflection of memory in rodents- it was not easy to show this conclusively). The first experiment shows a correlation between chemotaxis changes and translocation of daf-16 to the nucleus (as does their previous Cell paper). In neither paper are there the control experiments to show how strong this correlation is or that it is necessary for the behavioral change. The authors need to address these questions: Can memory exist in the absence of daf-16 translocation (here they could pre-screen animals with behavioral test prior to imaging)? Can daf-16 translocation happen when behavioral memory is not expressed (it appears as though it does in some mock trained animals)? Is a daf-16 mutant unable to show the leaning/memory? Reminder correlation is not causation!

2. Similarly, in figure 1 in the trained group there is a difference in chemotaxis just over 20%, however the daf-16 data show a difference of ~35%. In the previous Current Biology paper the behavioral effect was even larger (~60%) while the fraction of animals showing the translocation was significantly smaller), If the daf-16 translocation is memory why are the differences between trained and control not the same? If the authors “pre-selected” worms that show the behavioral effect to image the correlation should be close to 1 and if they chose worms that did not learn it should be closer to 0. This would strengthen their claims that daf-16 is a good proxy for memory.

3. It is concerning that F1 animals do not inherit IAA avoidance behavior (Figure 1F). Is this really associative memory inheritance or is it inheritance of “cellular memory” or inheritance of a more sensitive stress response? The term “associative memory” is typically used to describe an association between a stimulus and a conditioned behavior. While 50 percent of P0s did indeed display an association between IAA and stress indicated by their avoidance of IAA, this behavior was not inherited in the F1 generation. Thus, I question whether this can really be considered inherited associative memory. Are there any other cases in the literature of a transgenerational inheritance that skips a generation? Further, the authors did not include other tests to assess the other characteristics of associative memory, such as the extinction of the memory which would require repeated exposure of IAA without starvation prior to the challenge with IAA. In the supplemental material, the authors did include data from an experiment where IAA was paired with food and found no nuclear translocation of DAF-16 with subsequent IAA exposure. However, this supplemental data does not provide much insight into whether there is stimulus specificity as food is inherently a non-stress inducing stimulus. Better to pair another compound with starvation and test IAA to see if there is daf-16 translocation. Conversely, in animals trained to associate IAA with starvation (regular training procedure), does DAF-16 nuclear translocation occur when the trained worms are exposed to another neutral scent? Can you pair starvation with another scent and get the same DAF-16 translocation results?

4. Within the P0 population there are relatively low incidences of learning to associate IAA with stress as there’s only a 35 percent increase of P0s that exhibit DAF-16 translocation when exposed to IAA post training, and 45 percent of trained P0 animals avoided IAA. Further, only about 20 percent of F1s and F2s exhibited DAF-16 translocation upon IAA exposure. In the captions of supplementary figure 3, the authors state that the rate of inheritance is lower than the expected for the F2 generation. However, the authors make no further comments or speculations on why the low learning/memory

rates and inheritance rates occur.

5. For figure 1 panel E, why start the pre-selection process at the F1 to F2 generation? If a pre-selection process had to occur to get positive F2s, why not re-try collecting F1 data by pre-selecting progeny from positive P0s? This may increase the rate of inheritance.

6. For experiments described in figure 2 panels B and C, there should be control conditions where 1) animals do not undergo any starvation but did receive the odorant and 2) animals who did not undergo starvation and did not receive an odorant. While the authors state that 50 percent of worms typically survive a 3-5 hour heatshock at 37°C, this rate could differ depending on how much temperature variation there is within the 3-5 hour exposure as well as background mutations in the lab's strains (and other lab-specific environmental conditions). The need for these controls are supported by the wide variety in rates of survival for both trained and mock-trained P0s and F1 animals across replications (ranging from 10-75%). Further, the authors do not currently rule out the possibility that IAA exposure impacts the survival of wildtype animals. Therefore, the suggested controls are needed to claim survival rates were increased. Further, the authors do not comment on or speculate why there were increased survival rates of mock trained F1 in 4 experimental replications.

7. Very interesting and solid approach to determine whether the oocyte or sperm pass along the associative memory (figure 3A). Is it possible for the authors to determine which small RNAs are being passed along via the sperm or how the resulting histone methylation impacts chromatin structure?

8. For all mutant strains throughout the rest of the paper it is critical to show the behavioral data as well as the *daf-16* data.

9. Figure 3B. The authors state the *wdr-5.1* is not involved in memory inheritance, however, there is still a decrease in the amount of F1 worms who show DAF-16 translocation in comparison to P0s. While the *wdr-5.1* may not be essential for memory inheritance, it does seem to have a minor role.

10. Figure 3B The presence of DAF-16 nuclear translocation in trained P0 and F1 *hlp-2* mutants is much lower than the other mutant strains. The authors conclude that *hlp-2* is not required for memory inheritance, but *hlp-2* mutants had an ability to associate IAA with starvation/stress- which was passed on to the F1 generation. In figure 3C, we see that decreased DAF-16 nuclear translocation in *nrde-2* P0 mutants – which again was passed down to the F1 generation. While the authors do suggest these genes may be involved in learning to associate IAA with stress, no further inquiries are made into the role of these genes. Following up on the details of how this is working would make a stronger paper than suddenly switching briefly to neuropeptides and then switching again to serotonin,

11. Insufficient justification is given for why the authors investigated the role of neuropeptides. The authors investigate the role of neuropeptides by assessing P0 and F1 worms with mutant alleles of *egl-3* and *unc-31*, and conclude that neuropeptides act downstream of memory formation to enable inheritance. However, *egl-3* P0 trained animals display decreased DAF-16 nuclear translocation compared to *unc-31* mutant animals (Figure 3D), suggesting that neuropeptides may be involved in forming the association between IAA and stress. However the authors do not comment on this in the discussion. Further, while the authors did show neuropeptides are required for memory inheritance, many reasonable questions were not addressed. Which neuropeptide(s) is/are involved? Which receptors are being acted on? Rather than suddenly dropping neuropeptides and switching to serotonin further investigation into which neuron(s) are secreting the neuropeptide(s) would be interesting. A potential starting place would be to degrade or KO *egl-3* or *unc-31* in the AWC neurons. Further, the authors could investigate which tissues receive the secreted neuropeptide(s). In the proposed model, the authors seem to hypothesize that the secreted neuropeptides act on the gonads. What neuropeptide receptors are expressed on the gonads and at the very least discuss how this could be investigated.

12. While the authors were able to show that serotonin does induce DAF-16 nuclear translocation in the trained P0 and their F1 progeny (via optogenetic activation of serotonergic neurons or exogenous serotonin), the cellular mechanism downstream of serotonin release is not investigated. Therefore, there is very little difference between the findings presented in the manuscript and the findings of Eliezer et al. 2019 (<https://doi.org/10.1016/j.cub.2019.03.059>).

13. Were DAF-16 expression levels measured for the different conditions (P0, F1, F2 – trained, mock-trained for wild-type animals and for all mutant strains)? Does exposing animals of any of the strains to multiple long periods of starvation during development cause higher basal expression levels of DAF-16, and therefore these animals are more likely to have a lower threshold for stress response initiation? While both the trained and mock-trained groups are undergoing the same starvation conditions, it may explain why we see increases of DAF-16 translocation in all groups in supplementary figure 1.

Minor

1. For figure 1 panels B,C, and D clearer labeling is needed to identify which structures are being identified by the white arrows. Inclusion of a scale bar and anterior/posterior compass would further increase figure clarity.
2. For figure 1 panel E, the “F2 no pre-selection” bars should be on the same axis as the others conditions.
3. Figure 2C. What is the purpose of including the data for the hrde-1 KO if the data is uninterpretable?

A point-by-point response to the reviewer's comments

We thank the reviewers for the helpful constructive comments that have greatly helped us to improve the manuscript. We have addressed all the comments and added an extensive amount of new data that further support our findings. Specifically, we have added behavioral analyses of P0 and F1 generations for all mutants, performed memory extinction experiments, repeated the experiments while training in different larval stages, and performed the various requested controls (e.g., preselecting memory-positive animals, assaying DAF-16 levels, controlling by training on food or histamine only and more). We have also better explained the methods, extended the discussion as suggested, and reformatted the necessary figures. Below we provide our point-by-point responses.

Reviewer #1 (Remarks to the Author):

In this paper, Deshe and colleagues describe the inheritance of associative memory across two generations and elucidate part of the underlying molecular and cellular mechanisms in *C. elegans*. Using an established associative learning paradigm, they first show that associative memory of odor-starvation pairing during larval development is transmitted to the progeny for up to two generations. This inherited memory is restricted to a physiological conditioned stress response, assessed by the nuclear translocation of the DAF-16/FOXO transcription factor, whereas a behavioral conditioned response is not inherited. The authors show that memory reactivation, when presenting the conditioned stimulus (the odor isoamylalcohol) to the parental or F1 generation, protects the animals from heat stress. By testing mutants defective for methylation, small RNA pathways and neuropeptide signaling, they identify a number of mutants that are specifically defective in memory inheritance. The authors then investigate whether the cellular basis for associative memory as characterized previously in the parental animals (Eliezer et al., 2019) is similar to that of memory in the F1 generation. They show that the odor-sensing neuron AWCOFF and downstream serotonin signaling is indeed required for memory retrieval in both the P0 and F1 generations.

The authors have presented a clear story and the identification of an inherited associative memory is a valuable contribution to the study of learning and memory. The manuscript clearly describes the experimental details and is overall well-written. However, the mechanism of memory inheritance remains unclear and several concerns need to be addressed to substantiate the authors' conclusions.

Major comments:

1. The authors state that “the inherited memory endowed the progeny with a fitness advantage”. However, it is unclear whether the experiment shown in Figure 2 was a competition experiment between trained and control animals, which would assess a potential fitness advantage, or whether heat stress resistance was quantified in trained and mock-trained animals separately, which would not directly assess fitness. The authors should clarify this and adjust their statements accordingly.

In the revised manuscript, we have adjusted the statements accordingly and refrained from using the term 'fitness advantage'. Instead, throughout the paper, we use the term survival chances or survival rates.

We also state that the experiment (figure 4 in the revised manuscript) quantified survival chances of the two animal groups when grown separately.

In the methods, page 30, lines 714-715:

“Heat shock and survival assays quantified % survival of trained and mock trained animals held separately (and not by direct competition).”

Additionally, it is not clear why there is a large variation and much higher N number for the experiments with the F1 animals.

The large variation in the survival chances was due to day-to-day variability in the heat stress procedure to achieve ~50% death. This procedure is very sensitive to its duration: for example, on one experimental day, ~4 hrs of heat stress induced 50% death and on another day it induced ~80% or barely 20% death. Moreover, a mere 15 or 30 minutes difference in stress duration could result in either 20% or 80% survival. To overcome this, we included several heat stress durations within the same experimental day. Supplementary figure 5 details all heat stress durations used across all the experiments and the resulting survival chances.

We now state and explain this in the Methods section, page 30, lines 706-713:

“As the inflicted % death considerably varied across experimental days, we analyzed varying durations of heat shocks. Notably, we always compared survival rates between same-day trained and mock-trained animals, where both groups were incubated together for the exact same duration at 37 °C. The P0-generation worms were heat shocked for 4.5-5.5 hours while the F1-generation worms that were never starved were heat shocked for 3.5-4.5 hours (Supplementary figure 5). The extended heat shock period used for the P0 generation was due to their higher resistance to heat, presumably due to the intense stress (repeated starvations) that they underwent throughout development.”

Since in initial experiments we were most interested to study these effects in the F1 progeny, we did not assay the trained P0 animals. Only after several experimental repeats, we assayed both P0 and their F1 progeny. For completeness, we report all of the experimental results obtained, and hence, the higher number of experimental repeats for the F1s.

2. Several mutants were identified that are specifically defective in memory inheritance, leading the authors to state that H3K9/H3K36 methylations, NRDE-3 and neuropeptide secretion are required for the inheritance. However, only one mutant allele of these genes has been tested; thus, it cannot be excluded that other genetic changes (e.g. background mutations) may be responsible for the observed inheritance defects in these mutants. Additional evidence is needed to show that these genes are indeed required for memory inheritance.

We verified that all mutants used in this study were backcrossed at least 5 times (customary used to reduce chances of off target background mutations). Backcrossing with N2 WT animals was done either by the researchers that initially reported/generated the mutants, or by us in case <5 backcrosses were indicated in the original publications (or if obtained via the Mitani lab through the National Bio-Resource Project).

3. The authors identified a number of candidate genes that could be involved in transmitting associative memory to the progeny. Overall however, the mechanism underlying memory inheritance, including the molecular identity of the transmitted signal(s) and the cellular mechanisms, remains unclear. For example, the cellular focus of the identified genes has not been investigated. Are these players, e.g. neuropeptide signaling, required within the memory circuit including AWC and/or serotonergic neurons, or are other cells/tissues involved?

With no prior knowledge, the origin of the signaling neuropeptides could be in any of the neurons. Moreover, these neuropeptides may be acting exclusively in the parental generation, or in their progeny (or in both). Even more so, different combinations of the neuropeptides may be playing critical roles in the parents and in the offspring to support the inheritance of the associative memory. Addressing all these interesting, yet perplexing, possibilities requires a significant extra effort of testing many different mutants, possibly also generating conditional mutants in which production of the neuropeptides is impaired in either the P0 or the F1 generations, and then validating these findings by generating targeted rescue strains in which expression can be induced in either of the generations. We believe such efforts will merit a separate publication.

4. Supplementary figure 2 shows that exposure to IAA in the presence of food does not induce DAF-16 translocation. This is an important control in the associative memory paradigm. However, the experimental details on the training protocol and the duration of IAA exposure as compared to the IAA-starvation pairing seem to be missing. This information should be included in the manuscript, and IAA exposure should be comparable to the starvation training protocol. It would also be helpful to describe the results of this experiment in the main text.

We now provide a detailed protocol for this control experiment in the methods section (page 25, 577-586):

***“Training worms on food.** To control for the possibility that the mere exposure to IAA may induce DAF-16/FOXO nuclear translocation, we trained the worms to associate IAA while on food. Since animals trained on food develop faster than animals trained with intermittent starvations, we used a similar training protocol as shown in figure 1A, but adjusted the IAA exposure schedule to the faster growth rate. Specifically, 24 hours post hatching, we added IAA for 24 hours. Worms were then washed off the plates and incubated for 6 hours in the absence of IAA, followed by an additional exposure to IAA for 18 hours, a wash, and 6 hours of recovery in the absence of IAA. At the end of this last step, the worms had reached the young adulthood*

stage and were subsequently assayed for DAF-16/FOXO nuclear translocation following IAA challenge (Supplementary figure 2A)."

We also highlight in the main text the result of this control experiment (page 4, lines 96-97): *"Notably, worms trained to associate IAA with food did not exhibit odor-evoked DAF-16/FOXO translocation to cells' nuclei, nor did their F1 progeny (Supplementary figure 2A)."*

5. Odor-evoked memory reactivation triggered both behavioral and physiological responses in the trained P0 generation, but only the physiological stress response seems to be inherited in the F1 and F2 generations. For this physiological response, F1 animals had to be pre-selected to see an effect of memory reactivation in the F2 generation. Could inter-animal variability also mask inheritance of the behavioral response? Around 45% of P0 animals avoided IAA after training (Fig 1F), but the authors did not test behavioral responses of F1 animals directly descending from these animals showing the learned behavior.

We have now carefully followed this suggestion and indeed found that F1 progeny, descending from the trained P0 animals that showed the memory-induced withdrawal output, also exhibited this withdrawal behavioral response. However, and in contrast to the cellular/physiological response, the learned behavioral response did not pass on to the F2s. These new results now appear in figure 4A.

Furthermore, following these new findings using WT animals, we comprehensively analyzed the memory-evoked behavioral responses of P0 and F1s for all mutant strains (Figures 5B and 6B). All the results and the conclusions from these new experiments are detailed throughout the manuscript.

6. The figure legend of supplementary figure 1 states that the overall stress in the population without IAA exposure was assessed by quantifying the % of worms with nuclear DAF-16 before IAA challenge. This value was subtracted from the % of worms with nuclear DAF-16 after the IAA challenge, and the same subtracted values were used in all other figures. Based on this statement, it is unclear whether the overall stress in the population was assessed for each experiment independently, as basal stress levels may vary between different experimental repeats.

Indeed, we assessed the basal stress level independently in each of the experimental repeats. In each such repeat, we scored ~50 worms. We now explicitly explain this in the legend to supplementary figure 1, page 39, lines 925-929:

"These data provide the raw values from which figure 2A was compiled: For each group we subtracted the % of worms with nuclear DAF-16/FOXO before the challenge with IAA from the % of worms with nuclear DAF-16/FOXO after the challenge with IAA. This subtraction was performed independently for each experimental repeat."

And in the Methods section, we detail about this quantification procedure, page 27 , lines 639-643:

“Each experiment always included two groups of worms: trained and mock-trained groups. We first quantified the baseline stress levels in both groups by scoring the % animals that show nuclear DAF-16/FOXO before the challenge. These values were then subtracted from the stress levels quantified in the same groups following the IAA challenge. This subtraction was conducted for each experimental repeat independently and these results were then averaged.”

7. Figure 4B: Histamine-induced silencing of the AWC-OFF neuron blocks retrieval of the conditioned stress response in P0 and F1 animals. However, the authors did not test whether histamine treatment alone, in the absence of the histamine-gated chloride channel, affects DAF-16 localization. This control is crucial to ensure that the observed effect is due to silencing of AWC activity rather than histamine exposure.

We have now performed this important control experiment, which appears as a new supplementary figure 7. We found that histamine treatment, in worms not expressing the histamine-gated chloride channel, does not affect DAF-16/FOXO nuclear localization, and memory reactivation is intact in both the P0 and the F1 animals.

We also included this statement in the results section, on page 16, lines 366-367:

“Importantly, histamine by itself does not affect the capacity to reactivate the memory in the P0 nor in the F1 progeny animals (Supplementary figure 7).”

Minor comments:

1. Most experiments, except for Figure 1, were only done for the F1 generation (not F2s). In the methods section, the authors mention that the analyzed F1 progeny were still in the unfertilized germ state during training of the P0 animals. Thus, F1 animals were not exposed to the training conditions in the embryonic stage. This information is important for the correct interpretation of data in Figures 2-4, but it is not clear from the main figures. It would help to clarify this in the experimental scheme of Figure 1A.

We have now added this information to the legend of the figure 2A that describes the different F1s tested at various hatching time windows, page 8, lines 173-175:

“48 h denotes that the assayed F1s hatched 48 hrs post the recovery of the trained P0 animals. Otherwise, assays were performed on F1s that hatched 24 h following recovery of the P0 trained animals, thus, F1s were not exposed to the training conditions in their embryonic stage.”

2. The authors cite the famous experiment of Pavlov as an example of classical conditioning. The original studies describing these experiments, however, did not mention the use of a bell, even though this is commonly stated as such. It would be better to rephrase this example, f.i. as an association between food and a sound or auditory cue.

We have now corrected this sentence accordingly, page 2, lines 28-31:

“A classic example is the Pavlovian dogs who associated food (the unconditioned stimulus, US) with an auditory cue (conditioned stimulus, CS). Consequently, when exposed to this sound cue, these dogs started salivating in expectation for the associated food”

3. Lines 46 and 48: *Pseudomonas* should start with a capital letter and should be written in italic.

We have corrected accordingly.

4. Fig 1B-D lack scale bars.

We have now added scale bars to all panels (figure 1B-C).

5. Line 147-148: “Heat shock typically kills around 50% of the worm population.” Does this refer to well-fed N2 animals or animals after odor-starvation training? This is not clear from the manuscript, because Fig. 2 shows a survival rate ranging between 10% and 80%.

We aimed to inflict a heat shock that will result in ~40-60% death among the odor-starved animals. This was indeed ~average death across all experimental repeats, though it greatly varied across experiments as discussed in the response to comment #1 above.

We now explain this in the Methods section, page 30, lines 704-709:

“The worms were then transferred to OP 50-containing plates with no IAA and subjected to a 37 °C heat shock, a stress that was partially lethal to animals. As the inflicted % death considerably varied across experimental days, we analyzed varying durations of heat shocks. Notably, we always compared survival rates between same-day trained and mock-trained animals, where both groups were incubated together for the exact same duration at 37 °C.”

6. Line 175 and 231-232: naïve should be naive.

We have corrected accordingly.

7. Line 203: “that were shown to function in...” (in is missing)

We have corrected accordingly.

8. Lines 206 and 219: state that *Argonaute* and neuropeptide secretion mutants show intact WT levels of memory formation and retrieval. However, Fig. 3B-D do not include WT control animals, so these statements are not supported.

Based on the new results, we now state that the *Argonaute* mutant showed intact cellular and behavioral memory outputs, but not their F1 progeny. We explain that these results should be compared with WT data which are provided in figure 2A and 4A. On page 14, 306-310:

“P0-trained animals, defective in NRDE-3, an Argonaute responsible for transferring small RNAs from the cytoplasm to the nucleus in somatic cells⁵³, showed intact cellular and behavioral memory outputs, but their progeny failed to show these memory-evoked outputs (Figure 6A-B, and compare to WT results shown in figures 2A and 4A).”

Likewise for the neuropeptides, we state (page 15, 329-333):

“Notably, while the parental trained generation showed intact physiological and behavioral memory-evoked outputs, F1 worms showed no retrieval capacities, suggesting that neuropeptides are also involved in the transmission of the associative memory (unc-31 mutants were not analyzed in the behavioral assay since they hardly move, Figure 6C-D).”

9. Fig. 3C: mutants of nrde-2 seem to show even stronger stress responses in the F1 generation than the P0 animals. Is this difference statistically significant?

We ran the statistical test and found it to be non significant. Overall, this mutant strain showed lower (but similar) capacities for memory retrieval in P0 trained animals and in their F1 progeny. Data in the revised version are in figure 6A.

10. Line 259: “Remarkably, the mere light activation of AWC-OFF [...] sufficed to induce a rapid stress response”. The authors’ previous work already showed that the AWC-OFF neuron is necessary and sufficient to trigger the stress response, so this finding is not remarkable or unexpected, but in line with previous reports.

We have now rephrased the sentence, page 16, 359-362:

“The mere light activation of the AWC^{OFF} neuron, either in the trained P0 or in their F1 descendants, sufficed to induce the stress response, as indicated by the rapid translocation of DAF-16/FOXO to cells’ nuclei (Figure 7A).”

11. Lines 317-318: “a combination of [...] and neuropeptide secretion may mediate...”

We have corrected the sentence, pages 19, lines 425-426:

“A combination of small RNAs, histone modifications, and neuropeptides secretion mediates this process (Figure 8).”

12. Supplementary Table 1: unc-31 mutant is missing from the table.

The *unc-31* mutant (e928 allele) crossed to the DAF-16 reporter strain is provided in the strain list table as strain ZAS178 {zls356 [daf-16p::DAF-16a/b::GFP + rol-6(su1006)] x *unc-31*(e928)}.

Reviewer #2 (Remarks to the Author):

This paper builds on an aversive association assay previously published by the same lab (Reference 3 in the reviewed paper). In the previous work they detailed the learning assay. In this paper they show that the associative memory established in the P0 generation (previously published work) can be passed (at low levels) to the F1 and F2 generations. Furthermore, they show that the inheritance requires some histone methyltransferases and neuropeptide secretion. They went on to pinpoint differing requirements for AWCOFF and AWCON neurons in the signaling pathway. Given that these neurons had already been implicated in the pathway in their previous paper, these findings aren't quite as novel as they would otherwise be. The work will be of significance to the field, and most conclusions drawn are sound, but there are additional experiments that the authors could have performed that would make the paper more thorough.

The only conclusions that I believe are wrong are those relating to the roles of the various genes tested in the assay (Figure 3B-D). In Figure 3C line 241/2 "mutants defective in the Argonaute *hrde-1* showed impaired memory... thus precluding inference of their role in the inheritance". This is true, but is surely also true to a lesser degree for *nrde-2* and *hpl-2*. Please adjust your conclusions.

For *hpl-2* and *wdr-5.1*, we have now refined our conclusions, specifically in light of the newly added behavioral data demonstrating that these genes are actually important to mediate memory-evoked behavioral outputs (page 13, 287-292):

"H3K4 methyl-transferase (WDR-5.1) and the heterochromatin protein (HPL-2), which binds methylated histones, do not seem to play a major role in inheritance of cellular memory, as both, the trained P0 animals and their F1 descendants show memory retrieval capacities, though with lower efficiency (Figure 5A). In contrast, inheritance of the behavioral memory was impaired in both of these mutants, suggesting that these genes play a role in mediating this type of memory to the progeny (Figure 5B)."

And for *nrde-2*, we restated (page 14, lines 310-316):

*"In contrast, mutants, defective in NRDE-2, the nuclear dsRNA-induced RNAi factor responsible for maintenance of small RNAs transgenerational inheritance^{53,52}, showed lower capacities for retrieval of the cellular memory when compared to WT, though these lower levels were comparable in both the P0 and their F1 progeny (Figures 6A, 2A). Retrieval of the behavioral memory were intact in both the P0 and the F1 generations (Figure 6B), suggesting that while *nrde-2* may affect formation or retrieval of associative memories, it is not likely to play a major role in inheritance of such memories."*

For the *hrde-1*, page 14, lines 316-322:

“Mutants in the hrde-1 gene, a key nuclear Argonaute for transgenerational inheritance in C. elegans⁴⁹, were defective in processes of cellular memory already at the trained parental generation, thus precluding from assessing its involvement in inheritance of cellular memories (Figure 6A). Nevertheless, this gene may be involved in inheritance of behavioral outputs as the trained P0 generation withdrew from the conditioned stimulus IAA while their F1 progeny failed to withdraw (Figure 6B).”

Line 198-201 Is the wdr-5.1 F1 level significantly lower than WT? IF so, I think that argues that WDR-5/1 does play a role in memory inheritance, albeit not as strong a role as met-2 and set-25.

We checked and *wdr-5.1* is not significantly lower than WT: For both, WT and *wdr-5.1*, P0 levels are ~35% and F1 ~25% (comparing data for WT found in fig. 2A and for *wdr-5.1* in fig. 5A).

Similarly, (Lines 207-211) *nrde-2* is here described as having intact capabilities for memory retrieval in P0 and F1 but appears to have substantially lower memory in P0 compared to WT (not shown on the graph).

As mentioned above, we have now restated the description for *nrde-2* (page 14, lines 310-316).

Figure 3D again, *egl-1* mutants do look slightly defective at P0. Comparisons with wildtype should be performed.

We compared P0 levels of *egl-3* (now in figure 6C) with WT (figure 2A) but did not detect a significant difference.

Additional experiments that would improve the paper.

Figure 1E and Supplementary figure 3 deal with the inheritance of the associative memories. In figure 1E the authors mix data from selected parents (F2) with that from unselected parents (F1). They then in Supp Fig 3 infer what they think the inheritance rate is from P0-F1 without actually doing the experiment. It would be better if they actually selected P0 animals that display the learning to produce the F1 generation. Then they can accurately quantify inheritance properly. Doing this would also allow them to sort out the role of *hrde-1* and *hpl-2* in learning vs memory

We have now followed this suggestion and tested F1s selected from positive P0 animals (figure 2A in the revised version). There was no difference between randomly picking F1 progeny and selecting those whose P0 showed a memory response (denoted in graph as selected from +P0). In contrast, while randomly selecting F2s failed to detect memory-positive animals, selecting F2s from positive F1s identified memory-positive animals in fractions equivalent to those found in F1s. This suggests that memory may be fading away quickly, such that by the F3 generation, we could not observe memory-positive animals, even after pre-selecting positive F2s.

Also in Figure 1E I was confused as to how the P0 untrained animals exhibited a negative percentage of nuclear DAF-16, and what is the delta on the y axis referred to until I read the Figure legend in Supp figure 1. I recommend including some of that info in the figure legend for Figure 1.

We now better explain how a negative percentage may be obtained. We added the following explanation to the legend of (currently) figure 2A, page 8, lines 165-170:

“Trained and mock-trained animal groups were each scored before and after challenging the animals with the conditioned stimulus IAA. % of worms with nuclear DAF-16/FOXO before the challenge were subtracted from the % of worms with nuclear DAF-16/FOXO after the challenge (thus, presented as $\Delta\%$). Negative $\Delta\%$ values could arise if % of animals with nuclear DAF-16 following the IAA challenge were lower than the % before the challenge (see supplementary figure 1 for extended description and the raw data used to derive these values).”

Figure 3A and Supplemental figure 4. The finding that training only at the L4 stage does not allow for memory transmission to progeny is striking, but a bit unsatisfying. It would have been nice to see this extended – the training consists of 4 periods of starvation that occur throughout larval development. It would be illuminating to train populations for each of these training steps individually in order to determine which of the larval stages are important for establishing memory (especially since this is talked about in the discussion).

We have now followed this suggestion and trained the animals in either the L1, L3 or the L4 stages and assayed memory transmission (new supplementary figure 3). We find that none of these stage-specific training resulted in memory formation in the trained generation, nor in their descendants, with the exception of L4-trained animals, in which the trained animals formed the memory but it was not transmitted to the progeny.

These results are provided in the Results section, pages 4-5, lines 103-109:

*“To transmit the memory to the F1 progeny, P0-generation animals underwent four rounds of spaced training, one in each of the four larval stages (**Figure 1A**). Associating the conditioned stimulus IAA with starvation in either the L1, or the L3 larval stages did not lead to memory formation in the P0, nor in their F1 progeny (**Supplementary figure 3**). Training the P0 generation exclusively during the L4 stage formed robust memory traces in the P0 animals, in agreement with a previous report³, but these memory traces were not evident in the F1 progeny (**Supplementary figure 3**).”*

And in the Discussion section, pages 21-22, lines 484-489:

*“A single training cycle during the L1 or the L3 larval stages did not lead to memory formation in the P0, nor the F1, generations (**Supplementary figure 3**). Worms trained with a single cycle during the fourth larval stage formed associative memories that were not passed on to their progeny. Together, these results suggest that germ cells are modulated early on during gonad*

development, but that it requires several training cycles for transmitting the memory to the progeny.”

It is also important to note that only the offspring laid (but not hatched) during the first 24 hrs of P0 adulthood were assayed. It would be interesting to wash the adults to fresh plates and then collect the second 24 hr batch of offspring. Would the pattern of memory be different? Would memory then be observed in the offspring of L4-only trained animals?

We have followed this suggestion and assayed the offspring from the second 24 hours, that is, 48 hours post recovery from training. These progeny also retained memory activation capacities as shown in figure 2A (F1 48 h). Since L4-only trained animals did not show any evidence for memory transmittance, we did not assay that training paradigm for inheritance in the following 24 hrs.

Other issues

Line 58 I could be wrong, but I thought the number of neurons is now officially down to 300?

We are aware of the non-official communication that the two CAN cells may be considered as non neural cells, and hence the total count will be 300. However, we could not find a reference to cite this, so we opted to keep the current official number as 302. We will be happy to modify the text if such a citable reference appears.

Starvation training – how many of the worms survived this starvation procedure? That’s quite a high level of starvation, was there a severe population bottleneck that occurred? Similarly, how dramatically was the fertility affected by the starvation training? Could this have implications for memory and learning?

We observed a very small fraction of the worms that died following these starvation rounds, so population bottleneck is not likely to occur. Also, we did not witness a fertility effect as the number of progeny was similar to the one observed for naive (non-trained) animals.

Figure 1 B/C/D the images supplied are low resolution and low magnification, making it very difficult to see the nuclei that we are supposed to be able to see. Higher magnification inserts with better resolution should be supplied. Can the nuclei be identified in some other way also? DAPI or showing the brightfield/DIC image? As another note- the worms appear to have a dumpy phenotype, which is particularly evident in D. This is not mentioned. Why are they dumpy?

We now provide enlarged images with higher resolution and a scale bar (figure 1B-C). As evident by the new panels, the worms are not dumpy. They may have looked dumpy in the previous version due to the smaller panels in which mid body regions were primarily zoomed in to aid visualizing the nuclei.

Figure 2B why is there such a huge range in survival in the P0 generation? Ranging from only 10% surviving to 75% - is it to do with the range of time that the heat shock was performed? 4.5-5.5 hrs for P0 is actually quite a substantial range. If the authors plot length of heat shock vs survival do they see a trend? I appreciate that this isn't the point of the paper, but it seems that there is a broad trend that the difference between trained/mock worms is larger when the mortality rate is more severe and so perhaps the data would be tighter and more convincing with a smaller spread of mortality.

A similar question was also raised by reviewer #1 above. For convenience, we provide below the same explanation:

The large variation in the survival chances was due to day-to-day variability in the heat stress procedure to achieve ~50% death. This procedure is very sensitive to its duration: for example, on one experimental day, ~4 hrs of heat stress induced 50% death and on another day it induced ~80% or barely 20% death. Moreover, a mere 15 or 30 minutes difference in stress duration could result in either 20% or 80% survival. To overcome this, we included several heat stress durations within the same experimental day. Supplementary figure 5 details all heat stress durations used across all the experiments and the resulting survival chances.

We now state and explain this in the Methods section, page 30, lines 706-713:

“As the inflicted % death considerably varied across experimental days, we analyzed varying durations of heat shocks. Notably, we always compared survival rates between same-day trained and mock-trained animals, where both groups were incubated together for the exact same duration at 37 °C. The P0-generation worms were heat shocked for 4.5-5.5 hours while the F1-generation worms that were never starved were heat shocked for 3.5-4.5 hours (Supplementary figure 5). The extended heat shock period used for the P0 generation was due to their higher resistance to heat, presumably due to the intense stress (repeated starvations) that they underwent throughout development.”

Figure 3, all panels: what do the *** refer to? Which comparison is significant?

Figure 3 now appears in figures 5-6 with additional data. In these figures, and in all other relevant figures (2-3,7) in which we made similar comparisons, we now made sure to include the following statement in the corresponding legends:

“Significance comparisons are between trained and mock-trained animals”

Line 259/260 It is unclear whether the P0 and F1 worms were light activated, or if it was only the P0 worms. Wording should be fixed to clarify.

We have now reworded the text such that it is clear that both P0 and F1 worms were light activated, page 16, lines 359-362:

“The mere light activation of the AWC^{OFF} neuron, either in the trained P0 or in their F1 descendants, sufficed to induce the stress response, as indicated by the rapid translocation of DAF-16/FOXO to cells’ nuclei (Figure 7A).”

Reviewer #3 (Remarks to the Author):

In the manuscript “Inheritance of associative memories in *C. elegans* nematodes” Deshe N et al developed a paradigm for testing the inheritance of starvation memory in *C. elegans* by associating starvation with the odorant isoamyl alcohol. They demonstrate through some elegant work that this starvation induces a translocation of DAF-16 which can be transmitted for 2 generations. Furthermore this causes a heritable increase in heat stress survival and is mediated by a paternally transmitted signal which requires the putative H3K9 methyltransferases MET-2 and SET-25, the putative H3K36 methyltransferase MET-1, the argonaute NRDE-3, as well as the neuropeptide signaling pathways involving EGL-3 and UNC-31. Finally they perform some experiments using different mechanisms of activating specific AWC neurons to demonstrate that these also have intergenerational effects on DAF-16 localization in the particular cells. All in all I thought this was an excellent and novel manuscript and while more work could be done to demonstrate a deeper mechanistic understanding of this phenomena I’m not sure if that is really required for this initial story. I think that most of the changes that are required are the addition of controls and altering the presentation of the data to more accurately reflect the findings.

Major Points

1) The majority of this manuscript hinges on DAF-16 localization. Since many treatments will cause DAF-16 localization changes the authors should be exceptionally careful about drawing a direct line between each of them. For instance the AWC results could be an independent mechanism from the starvation results. Since this is the only proxy for the phenomena if an additional metric could be added to bolster the findings that would greatly strengthen the findings.

In the revised version, we have added the memory-evoked behavioral outputs as an additional metric to the DAF-16 cellular/physiological memory readout. We found that memory-induced behavioral outputs are not necessarily linked to the physiological/cellular ones (as shown in supplementary fig. 6). Nevertheless, in the revised version, we attempted to be more careful drawing a direct link between DAF-16 localization and memory.

We believe that the experiments targeting AWC involvement in memory formation and reactivation are well controlled for other possibilities to be involved. For example, the mock controlled animals went through starvation in the absence of the conditioned odorant IAA and they do not show AWC-mediated memory reactivation. In addition, we have now added a control showing that animals exposed to IAA while one food do not form the stressful

associative memory (supplementary figure 2A), further underscoring the possibility that AWC neurons are involved in memory formation and reactivation following the association between starvation and IAA.

Additionally it is important to represent all these plots as dot plots with each individual experiment represented by a different dot on the bar graph so the readers can judge the magnitude and consistency of the phenotype. I would also include numbers in the text so that readers can judge the magnitude of the phenotype in addition to visualizing them.

Indeed. We have now replotted all the graphs to include dot plots that reflect the results from individual experimental repeats. Where possible, and to avoid word cluttering due to the many conditions and controls, we included numbers in the text.

Minor Points

1) C. elegans are nematodes so it is redundant to have both in the title

We have now removed 'nematodes' from the title.

.2) There are more than 3 mechanisms that could underlie transmittance of epigenetic information. Beyond DNA methylation, chromatin modifications and small RNAs there are also microbiota, mitochondria or maternally inherited proteins, prions... Those 3 have been the best studied but there is no indication that they are the main three.

We have now rephrased the sentence accordingly, page 12, lines 269-271:

"In C. elegans, small RNAs and histone modifications are the best characterized epigenetic mechanisms underlying transgenerational inheritance^{29,25,31,27,23,17}."

3) Fig 1 B-D needs to be labeled on the figure itself what are the conditions and what is being assessed should all be visible in the figure. For 1E (and all of the subsequent plots) it would be best to do a dot plot to represent each of the individual experiments so readers can visualize the magnitude of the differences rather than just representing the mean +/-SEM.

In the revised version, these panels are now fig 1 B-C. They are labeled to reflect the generation shown and the corresponding trained or mock trained animal groups.

We have also modified all figures to include dot plots.

4) As a negative control the authors should show that the odor itself does not alter daf-16 localization.

We have performed this control experiment and the results are shown in supplementary figure 2 (discussed above in point #1) . Indeed, the odor itself does not alter DAF-16 localization.

5) The color in Fig 2B+C is unnecessary and distracting. The authors should represent the heat stress survival data as a traditional bar plot with dots representing each experiment and

include statistics. Additionally in this section it would be good to cite the work that has shown that starvation in the parents leads to increased heat stress survival in the descendants (Jobson et al 2015, Webster et al 2018).

We have now changed the figure to a dots plot representing each experiment and included the statistics (figure 4C in this new version).

We also added the related references, page 10 lines 237-238 :

“Starvation alone of the parental generation was shown to enhance survival of the progeny when facing a heat stress^{12,44}. “

6) It is probably worth mentioning that met-2, set-25, and wdr-5 have all been implicated previously in inheritance of non-genetic information in *C. elegans* and including the appropriate citations.

We have now added the appropriate citations for these genes (refs 30, 45-48). On page 12, lines 276-277:

*“First, we studied key histone modulators, previously implicated in mediating epigenetic inheritance in *C. elegans* animals^{30,45-48},”*

And also for the corresponding small RNA pathways genes (refs 24, 25, 49-52), page , lines :
*“Next, we analyzed key genes in the small RNA pathways that were shown to function in *C. elegans* transgenerational inheritance^{24,25,49-52}. ”*

7) It is a good idea to cite the papers demonstrating that unc-31 and egl-3 are defective in neuropeptide secretion pathways when describing that you are testing those genes.

We have now added these citations (refs 55-56), on pages 14-15 , lines 327-329:

“We therefore analyzed memory inheritance in two different mutants ,unc-31 and egl-3, that are defective in neuropeptide secretion pathways^{55,56}. ”

8) In the discussion it is important to state that these conclusions are based on correlative experiments. The fact that genes involved in H3K9, H3K36, small RNA inheritance, and neuropeptides are important for the phenotype does not ensure that those molecules themselves are causal. I think if the authors wish to state causality a lot more work would be required. This is not necessary for this manuscript but a tempering of the conclusions is.

Agreed. We now state in the discussion that the involvement of these genes in transmittance of associative memories is correlative rather than causative, pages 20- 21, lines 457-460 :

“Nevertheless, our experiments using the various mutant strains (histone modifiers, small RNA-related genes and neuro-transmitter/peptide secretion factors) provide merely correlative, rather than causative, indication for their involvement in transmittance of associative memories.”

Reviewer #4 (Remarks to the Author):

In this study the authors investigate whether aversive associative memory can be inherited using the *C. elegans* model system. The unconditioned stimulus was starvation and the conditioned stimulus was isoamyl alcohol (IAA) – a normally attractive odorant. The training protocol for P0 animals consisted of inducing 4 periods of starvation at each larval stage (beginning at the L1) in the presence of IAA which lasted anywhere from 12-64 hours. Starvation periods were followed by a 6-24 hour recovery period in which the animals were on food and not exposed to IAA. The authors used translocation of GFP-tagged FOXO/DAF-16 into the nucleus of gonad sheath cells as an indicator of stress response activation. Trained P0 animals showed DAF-16 nuclear translocation after the mere exposure to IAA post training whereas mock-trained control animals did not. The F1 and F2 progeny of the trained P0 worms also showed DAF-16 nuclear translocation when exposed to IAA without ever undergoing the training protocol. In trained P0 animals and their F1 progeny, the induced stress response from IAA exposure increased survival rates following heat shock. The authors then investigate mechanisms mediating the transgenerational inheritance of the aversive associative memory and found that the memory is passed via the sperm. By training and testing animals and their F1 progeny that carrying knockout alleles of genes involved in small RNA pathways and chromatin modification processes, the authors determine that methylation of H3K9 and H3K36 as well as the NRDE-3 gene are important for the inheritance of the associative memory. The authors then determine that genes involved in neuropeptide synthesis and release are also important for memory inheritance. Using optogenetics and a histamine-gated chloride channel, the authors determine that the aversive associative memory is encoded by the AWC neurons, and activation of the AWCo neuron in the F1 progeny of trained worms could induce DAF-16 nuclear translocation. Optogenetic activation of serotonergic neurons in trained P0 animals and their F1 progeny, or addition of exogenous serotonin, could induce nuclear translocation of DAF-16, suggesting serotonin plays a role in the activation of the aversive associative memory. The authors conclude that

Summary

- What are the major claims of the paper?

This paper claims that, in *C. elegans*, aversive associative memories can be inherited to provide offspring with a fitness advantage through activating the stress response. The authors use mutant analysis, optogenetics, and other genetic techniques to indicate the inheritance of the physiological response memory is passed via the sperm, relies of neuropeptides, and genes involved in chromatin modification and small RNA pathways – whereas the associative memory is encoded by AWC off neuron and optogenetic activation of this single neuron or serotonergic neurons, or exogenous application of serotonin, can activate the inherited physiological response memory in the progeny.

- Are the claims novel? If not, please identify the major papers that compromise novelty
As the authors state, previous literature has indicated that transgenerational inherited of different forms in different model systems. Inheritance of other types of memories have been shown to be inherited via sperm (<https://www.biorxiv.org/content/10.1101/2020.11.18.389387v2>).

The genes chosen for mutant analysis have well-known roles in key pathways involved in epigenetic inheritance. While the authors were able to conclude which of these genes are important for the inheritance of the physiological response memory, no novel genes or pathways were investigated. Further, many additional lines of inquiry such as which neurons are involved and serotonin's involvement in stress response induction have been explored in the lab's previous publication regarding aversive associative memory (<https://doi.org/10.1016/j.cub.2019.03.059>). Therefore, the main contribution of this paper is confirming whether these genes/mechanisms are also involved in inheriting and or inducing the physiological response induced by the aversive associative memory tested in the F1 generation.

We now discuss our data in light of previous reports and reference the new manuscript mentioned above. More details are in the point-by-point responses below.

- Will the paper be of interest to others in the field? Yes
- Will the paper influence thinking in the field? If correct controls are run.
- Are the claims convincing? If not, what further evidence is needed?

Key control experiments are missing and many reasonable lines of inquiry were not addressed.

We have now added all the key control experiments as suggested in the point-by-point list below.

- Are there other experiments that would strengthen the paper further? How much would they improve it, and how difficult are they likely to be?

See comments below.

- Are the claims appropriately discussed in the context of previous literature?

The authors need to more clearly address how their work relates to their own previous publication and the work of other major *C. elegans* researchers (such as Dr. Oded Rechavi, Dr. Coleen Murphy, and Dr. Catharine Rankin).

We now discuss how our work relates to previous publications in the field.

- If the manuscript is unacceptable in its present form, does the study seem sufficiently promising that the authors should be encouraged to consider a resubmission in the future? Yes – however they need to add behavioral data for all mutant lines tested, not just *daf-16* translocations as described below.

We have now added behavioral data for all mutant lines.

Is the manuscript clearly written? If not, how could it be made more accessible?

The manuscript is written very clearly and the experiments are well described in the methods section. There are only a few minor comments about figure formatting or labeling below.

We have now also reformatted the figures and their labels accordingly.

- Could the manuscript be shortened to aid communication of the most important findings?

No, if anything the authors need to provide more information to support their claims.

Indeed, the revised version provides extensive new supporting evidence.

- Have the authors done themselves justice without overselling their claims?

The authors are overselling their claims, as there was no inheritance of a behavior, only a physiological response. Further, while the authors were able to show the involvement of genes involved in epigenetic changes, neuropeptide production/secretion and serotonin release were involved in memory re-call/inheritance, the authors provide very little novel evidence of the underlying mechanisms being affected/activated.

We now performed the required experiments and show that the behavior is also inherited.

- Have they been fair in their treatment of previous literature?

The authors probably could provide more information about the findings of previous literature on inheritance of other forms of memory or physiological responses. Further, it seems strange that work from major *C. elegans* researchers who focus on memory or transgenerational inheritance were not examined in paper's discussion. Are there any major mechanistic differences between the inheritance of associative versus non-associative memories?

We now discuss the relevant literature and explicitly emphasize that a similar set of genes is involved in associative and non-associative inheritance processes. Moreover, our findings indicate that memory-evoked cellular and behavioral outputs may be decoupled (detailed in the point-by-point response).

Have they provided sufficient methodological detail that the experiments could be reproduced?
The methods section provided sufficient detail to allow others to replicate their procedures.

Is the statistical analysis of the data sound?

Each experiment usually included multiple experimental replicates which scored 50 worms each. Were stats done on individual replicates or on cumulative data? If cumulative did each replicate give the same results?

Stats were done on replicates (e.g., the mean \pm sem is of 3-5 experimental repeats).

The authors chose to use the proportion test for their analysis but do not give a rationale for the choice. From <https://www.itl.nist.gov/div898/handbook/prc/section2/prc24.htm>: Because the test is approximate, N needs to be large for the test to be valid. What is the N used for each analysis (is the N per group or the overall N for all of the animals compared). Does the analysis take into account multiple tests on a single experiment, ie for figure 3B there are 10 groups (5 comparisons) that must have been tested- how is this accounted for?

The N is large enough since it is for the overall animals compared. Each statistical test compares the proportion of trained and mock-trained animals (for a particular genotype) rather than to WT.

- Should the authors be asked to provide further data or methodological information to help others replicate their work? (Such data might include source code for modelling studies, detailed protocols or mathematical derivations).
- Are there any special ethical concerns arising from the use of animals or human subjects? No

Most of the points mentioned above are also raised in the specific comments below. We provide a detailed point-by-point response in the following sections.

Major comments

1. Memory is a behavioral construct, not a physiological process (those would be described as cellular mechanisms underlying memory or cellular correlates of memory)- without behavioral data at best this might be a correlate of memory- not the memory itself- the authors lose track of this and consider the daf-16 to be the memory. (It took almost 10 years to convince people that LTP was a real reflection of memory in rodents- it was not easy to show this conclusively). The first experiment shows a correlation between chemotaxis changes and translocation of daf-16 to the nucleus (as does their previous Cell paper). In neither paper are there the control experiments to show how strong this correlation is or that it is necessary for the behavioral change. The authors need to address these questions: Can memory exist in the absence of daf-16 translocation (here they could pre-screen animals with behavioral test prior to imaging)? Can daf-16 translocation happen when behavioral memory is not expressed (it appears as though it does in some mock trained animals)? Is a daf-16 mutant unable to show the leaning/memory? Reminder correlation is not causation!

We have now followed this comment and performed comprehensive behavioral assays to all mutant strains. These memory-evoked behavioral output data now complement all the memory-evoked cellular/physiological memory outputs, as reflected by DAF-16 translocation to cells' nuclei. These new behavioral data are now presented in figures 4A, 5B and 6B, and the corresponding results are detailed throughout the text.

Furthermore, we performed the suggested experiment where we prescreened the animals with a behavioral test and then imaged them to assay reactivation of the cellular memory (DAF-16). These results are presented in the new supplementary figure 4. We find that behavioral (withdrawal) memory can exist in the absence of cellular memory, and vice versa, where non-withdrawing animals can still show memory-evoked cellular stress responses. Thus, these findings suggest that the two memory constructs are not necessarily coupled (page 10, lines 224-228:

“The odor-evoked withdrawal response and the odor-evoked cellular/physiologic response were not necessarily coupled: ~20% of the animals that withdrew from IAA subsequently exhibited a cellular stress response following exposure to IAA. A similar ~20% fraction of the animals that

did not withdraw from IAA, later showed odor-evoked stress responses (Supplementary figure 4)."

We did not test memory inheritance in *daf-16* mutants as we used DAF-16 as a readout for the cellular memory. We make no statements regarding DAF-16 involvement in this process. We also made sure to state that the use of mutants provides correlative rather than causative explanations, page 20-21 lines 457-460:

"Nevertheless, our experiments using the various mutant strains (histone modifiers, small RNA-related genes and neuro-transmitter/peptide secretion factors) provide merely correlative, rather than causative, indication for their involvement in transmittance of associative memories."

2. Similarly, in figure 1 in the trained group there is a difference in chemotaxis just over 20%, however the *daf-16* data show a difference of ~35%. In the previous Current Biology paper the behavioral effect was even larger (~60% while the fraction of animals showing the translocation was significantly smaller), If the *daf-16* translocation is memory why are the differences between trained and control not the same? If the authors "pre-selected" worms that show the behavioral effect to image the correlation should be close to 1 and if they chose worms that did not learn it should be closer to 0. This would strengthen their claims that *daf-16* is a good proxy for memory.

Following our new comprehensive analysis of memory-reactivated behavioral outputs, we concluded that though both memories are significantly evident in the trained P0 animals and their descendents, the cellular/behavioral memory is not coupled to the behavioral memory (as detailed in the previous point).

The difference in the behavioral results between the current study and the previous Current Biology manuscript lies in the difference in the training paradigms. In the previous manuscript, we studied the memory within the parental trained generation, where the training procedure consisted of a single cycle of coupling starvation + odor at the L4 stage only. This paradigm, however, did not lead to memory inheritance as shown in the supplementary fig. 3. To induce memory inheritance we modified the training paradigm to include four starvation cycles coupled with the conditioned odorant as shown in fig. 1A. These differences in the training paradigms lead to different output magnitudes following memory reactivation.

3. It is concerning that F1 animals do not inherit IAA avoidance behavior (Figure 1F). Is this really associative memory inheritance or is it inheritance of "cellular memory" or inheritance of a more sensitive stress response? The term "associative memory" is typically used to describe an association between a stimulus and a conditioned behavior. While 50 percent of P0s did indeed display an association between IAA and stress indicated by their avoidance of IAA, this behavior was not inherited in the F1 generation. Thus, I question whether this can really be considered inherited associative memory. Are there any other cases in the literature of a transgenerational inheritance that skips a generation?

We have now repeated the behavioral experiments, this time also assaying F1s that were pre-selected from avoiding (withdrawing) P0 trained animals (Figure 4A). We find that these

preselected animals do indeed avoid the conditioned stimulus IAA, thus showing the classical memory-induced behavioral outputs. Compared with the memory-induced cellular outputs, these behavioral outputs show weaker effects, as evident already in the trained P0 generation animals. These memory-induced behavioral outputs are not observed in the F2s, even if the assayed F2s were pre-selected from F1s who showed the withdrawal behavior.

Further, the authors did not include other tests to assess the other characteristics of associative memory, such as the extinction of the memory which would require repeated exposure of IAA without starvation prior to the challenge with IAA.

We have now performed the suggested experiment to assess extinction of the cellular memory. The results indicated that the memories in both the P0 trained animals and in their F1 progeny are robust and that they cannot be easily extinct. These data are now presented in the new figure 3, and the results are described on page 8, lines 187-195:

“We found that these associative aversive memories were very stable that could not be easily extinct (Figure 3). In the P0-generation animals, odor-evoked memory reactivation did not lead to nuclear translocation of DAF-16/FOXO only by the third extinction attempt. Interestingly, while these P0-trained animals lost their memory reactivation capacity, their F1 progeny retained this capacity. Similarly, extinction of the memory in the F1 progeny was possible if these descendants were subjected to two extinction rounds (Figure 3). Thus, the memory transmitted via the sperm is stable, and even when the memory is extinct in the trained parental generation, the information is successfully transferred to the progeny.”

In the supplemental material, the authors did include data from an experiment where IAA was paired with food and found no nuclear translocation of DAF-16 with subsequent IAA exposure. However, this supplemental data does not provide much insight into whether there is stimulus specificity as food is inherently a non-stress inducing stimulus. Better to pair another compound with starvation and test IAA to see if there is daf-16 translocation.

These experiments were actually performed in our previous manuscript (Eliezer et al, Curr Biology 2019). We paired additional odorants (e.g. diacetyl) with starvation and then tried to evoke the memory with IAA, and vice versa, train with IAA and evoke the memory with other odorants. We found that memory reactivation is neuron specific; that is, if IAA is sensed primarily by the AWC neurons, then other odorants, sensed primarily by the AWC neurons, can equally evoke the memory. Diacetyl, for example, whose primary sensory neuron is AWA, failed to evoke the memory (DAF-16 nuclear translocation).

Conversely, in animals trained to associate IAA with starvation (regular training procedure), does DAF-16 nuclear translocation occur when the trained worms are exposed to another neutral scent? Can you pair starvation with another scent and get the same DAF-16 translocation results?

Yes, as explained above, it is possible to pair starvation with one scent and evoke the memory (DAF-16 translocation) with a different scent as long as these scents are sensed by the same sensory neurons. Specifically, we have now addressed this question and studied whether this is the case for both the P0 trained animals and their F1 progeny: We trained animals by coupling IAA with starvation and were able to successfully evoke the memory using benzaldehyde, an odorant known to be sensed by the AWC neurons as well (results are shown in fig 7C). These results confirm that the memory is cell specific rather than scent specific in both the P0s and their F1s progeny. The results are provided on pages 16-17, lines 371-376:

“The possibility to induce memory retrieval by light activating specific neurons suggests that the memory is cell specific rather than stimulus specific. In this case, other stimuli, sensed by the AWC neurons (e.g., benzaldehyde), may be equally used for memory retrieval training with IAA as the conditioned stimulus. Indeed, using benzaldehyde, the associative stressful memory could be successfully reactivated in animals trained to associate IAA with starvation, in both the trained P0 generation and in their F1 progeny (Figure 7C).”

4. Within the P0 population there are relatively low incidences of learning to associate IAA with stress as there's only a 35 percent increase of P0s that exhibit DAF-16 translocation when exposed to IAA post training, and 45 percent of trained P0 animals avoided IAA. Further, only about 20 percent of F1s and F2s exhibited DAF-16 translocation upon IAA exposure. In the captions of supplementary figure 3, the authors state that the rate of inheritance is lower than the expected for the F2 generation. However, the authors make no further comments or speculations on why the low learning/memory rates and inheritance rates occur.

In the current modified version of the manuscript, we have comprehensively studied the inheritance rate. Specifically, and as suggested above, we have assayed pre-selected progeny whose parental generation showed behavioral (avoidance) or cellular (DAF-16 translocation) memory outputs. These experiments showed that the cellular memory can be observed up to (including) the F2 generation, but not in the F3 generation. The rate of this memory is reduced from >30% in the P0 generation to ~20% in the F1 and F2 generations (fig 2A).

Regarding the memory-induced behavioral output, this memory is observed in the F1 only (fig 4A). Thus, it is clear that the inheritance rates decrease with generations and in the discussion we speculate that this could be due to active mechanisms that limit such processes to few generations only, on page 22 lines 511-513 :

“In that respect, it may be interesting if inheritance of associative memories, like other transgenerational processes, is actively regulated to be limited to a few generations only⁶⁷.”

5. For figure 1 panel E, why start the pre-selection process at the F1 to F2 generation? If a pre-selection process had to occur to get positive F2s, why not re-try collecting F1 data by pre-selecting progeny from positive P0s? This may increase the rate of inheritance.

We have now followed this suggestion and repeated the experiment by pre-selecting positive P0 animals (figure 2A in the revised version). We find that the % of memory-positive pre-selected F1 animals is similar to non pre-selected F1 animals.

Similarly, we performed the same pre-selection of P0 positive animals that showed the behavioral output of withdrawal (Fig 4A). Again, the % of F1 avoiding (withdrawing) was similar whether the animals were pre selected or not, though this memory capacity did not pass on to the F2s, even when assaying F2s that were pre-selected from memory-positive F1s.

6. For experiments described in figure 2 panels B and C, there should be control conditions where 1) animals do not undergo any starvation but did receive the odorant and 2) animals who did not undergo starvation and did not receive an odorant. While the authors state that 50 percent of worms typically survive a 3-5 hour heatshock at 37°C, this rate could differ depending on how much temperature variation there is within the 3-5 hour exposure as well as background mutations in the lab's strains (and other lab-specific environmental conditions). The need for these controls are supported by the wide variety in rates of survival for both trained and mock-trained P0s and F1 animals across replications (ranging from 10-75%). Further, the authors do not currently rule out the possibility that IAA exposure impacts the survival of wildtype animals. Therefore, the suggested controls are needed to claim survival rates were increased. Further, the authors do not comment on or speculate why there were increased survival rates of mock trained F1 in 4 experimental replications.

We have now performed the suggested control experiment where animals underwent starvation while being exposed to the odorant IAA. The results (appear in the new supplementary figure 6) indicate that these animals do not show higher survival rates when compared to mock trained animals that were grown on food in the absence of IAA (that's actually the second suggested control experiment mentioned above). Indeed, an additional control experiment (supplementary figure 2) showed that training with IAA in the presence of food does not lead to odor-evoked nuclear translocation of DAF-16.

To control for possible temperature variations during the 3-5 hours of heat stress, we made sure to make pairwise comparisons between trained and mock trained animals that were trained side by side, and which following the training, underwent exactly the same procedure by spending exactly the same amount of time in the heat chamber. Moreover, since we observed a considerable day-to day variability in survival chances, we assayed and compared pairwise matched (trained and mock-controls) groups that spent different time durations in the heat chamber. These time durations and the paired-matched survival results between trained and mock trained animals are shown in supplementary fig. 5.

Notably, possible background mutations cannot explain the observed results. These experiments and thereof repeats lasted in the lab for > 1 year during which time we replaced the WT N2 animals at least three times, each time starting off with a newly-thawed batch of worms frozen years ago, thus being close as possible to the original WTstrain. On each experimental day, we started with a homogenous synchronized population of thousands of worms that was split to the two groups: trained and mock trained. Basing our results on such big cohorts that

were randomly split to two is not likely to produce such differences. Thus, we believe that our experimental design rules out such technical 'noise'.

We believe that the four experiments in which the mock trained animals show slightly higher survival chances in the F1s fall within the experimental noise, and hence the replicates that verify the significance of the phenomenon.

7. Very interesting and solid approach to determine whether the oocyte or sperm pass along the associative memory (figure 3A). Is it possible for the authors to determine which small RNAs are being passed along via the sperm or how the resulting histone methylation impacts chromatin structure?

We are also very interested in revealing the identity of the small RNA molecules. We have been working to this end in the past two years and hopefully will be able to publish another comprehensive paper describing the nature of these small RNAs.

8. For all mutant strains throughout the rest of the paper it is critical to show the behavioral data as well as the *daf-16* data.

We have now performed extensive behavioral assays for the WT and all mutant strains. These data are now provided in figures 5-6.

9. Figure 3B. The authors state the *wdr-5.1* is not involved in memory inheritance, however, there is still a decrease in the amount of F1 worms who show DAF-16 translocation in comparison to P0s. While the *wdr-5.1* may not be essential for memory inheritance, it does seem to have a minor role.

We agree and we therefore restated that *wdr-5.1* does not seem to play a major role in mediating the inheritance of the cellular memory. However, following our new behavioral experiments it seems that this gene controls the inheritance of the behavioral memory (fig. 5B). page 13, lines 287-292:

"H3K4 methyl-transferase (WDR-5.1) and the heterochromatin protein (HPL-2), which binds methylated histones, do not seem to play a major role in inheritance of cellular memory, as both, the trained P0 animals and their F1 descendants show memory retrieval capacities, though with lower efficiency (Figure 5A). In contrast, inheritance of the behavioral memory was impaired in both of these mutants, suggesting that these genes play a role in mediating this type of memory to the progeny (Figure 5B)."

10. Figure 3B The presence of DAF-16 nuclear translocation in trained P0 and F1 *hlp-2* mutants is much lower than the other mutant strains. The authors conclude that *hlp-2* is not required for memory inheritance, but *hpl-2* mutants had an ability to associate IAA with starvation/stress- which was passed on to the F1 generation. In figure 3C, we see that

decreased DAF-16 nuclear translocation in *nrde-2* P0 mutants – which again was passed down to the F1 generation. While the authors do suggest these genes may be involved in learning to associate IAA with stress, no further inquiries are made into the role of these genes. Following up on the details of how this is working would make a stronger paper than suddenly switching briefly to neuropeptides and then switching again to serotonin,

The rationale behind the ‘mini-screen’ using the different epigenetic mutant strains was to characterize which genes participate in inheritance of associative memories. Could these be the same histone modification and small RNA pathways that were shown to mediate inheritance of experience in different environmental conditions? Here we showed that indeed some of the known genes play a role while others do not (or that they have minor roles).

We agree that deciphering their mechanistic roles would be valuable, though this would also warrant a separate dedicated study(ies). We hope to include such studies in the future, coupled with our current efforts to reveal small RNAs that are transferred across generations. Together with future RNA and ChIP-seq analyses, we believe that such experiments will shed light on the molecular mechanisms by which associative memories are inherited.

11. Insufficient justification is given for why the authors investigated the role of neuropeptides. The authors investigate the role of neuropeptides by assessing P0 and F1 worms with mutant alleles of *egl-3* and *unc-31*, and conclude that neuropeptides act downstream of memory formation to enable inheritance. However, *egl-3* P0 trained animals display decreased DAF-16 nuclear translocation compared to *unc-31* mutant animals (Figure 3D), suggesting that neuropeptides may be involved in forming the association between IAA and stress. However the authors do not comment on this in the discussion.

We now provided in the text the rationale for analyzing the involvement of neuropeptides. On page 14, lines 324-327:

“We also speculated that neuropeptide signals may be at play in transmittance of stressful associative memories. This is because neuropeptides are known to relay nutritional status to the germline, and act non-autonomously upstream of the Insulin/IGF signaling pathway to regulate DAF-16-dependent germline proliferation⁵⁴.”

Further, while the authors did show neuropeptides are required for memory inheritance, many reasonable questions were not addressed. Which neuropeptide(s) is/are involved? Which receptors are being acted on? Rather than suddenly dropping neuropeptides and switching to serotonin further investigation into which neuron(s) are secreting the neuropeptide(s) would be interesting. A potential starting place would be to degrade or KO *egl-3* or *unc-31* in the AWC neurons. Further, the authors could investigate which tissues receive the secreted neuropeptide(s). In the proposed model, the authors seem to hypothesize that the secreted neuropeptides act on the gonads. What neuropeptide receptors are expressed on the gonads and at the very least discuss how this could be investigated.

Extracting which of the ~200 neuropeptides and their cognate receptors on the relevant neurons are involved in the inheritance requires additional massive work that we believe is well beyond the scope presented herein. While these are indeed interesting questions that we plan to delve into, we anticipate that such efforts will require a few more years of intensive research.

We now discuss the possible involvement of neuropeptides in mediating inheritance of the associative memory, page 20, lines 450-455:

“An intriguing possibility is that the inherited associative memory is established by integrating two seemingly unrelated cues: starvation that is signaled via selective neuropeptides, and the olfactory cue that is encoded by specific small RNAs targeting specific chromatin regions. Indeed, both neuropeptides and small RNAs were shown to be transferred from somatic tissues to the germline to affect the progeny^{39,62,40,41,22}.”

12. While the authors were able to show that serotonin does induce DAF-16 nuclear translocation in the trained P0 and their F1 progeny (via optogenetic activation of serotonergic neurons or exogenous serotonin), the cellular mechanism downstream of serotonin release is not investigated. Therefore, there is very little difference between the findings presented in the manuscript and the findings of Eliezer et al. 2019 (<https://doi.org/10.1016/j.cub.2019.03.059>).

This is correct. The purpose of these experiments was to study whether serotonin is involved in inducing the systemic stress response in the F1s, just like in the trained parental generation. Our findings show that indeed this mechanism is shared between the parental trained animals and their offspring.

13. Were DAF-16 expression levels measured for the different conditions (P0, F1, F2 – trained, mock-trained for wild-type animals and for all mutant strains)? Does exposing animals of any of the strains to multiple long periods of starvation during development cause higher basal expression levels of DAF-16, and therefore these animals are more likely to have a lower threshold for stress response initiation?

We have now addressed this possibility and quantified expression levels of DAF-16. We did this by quantifying fluorescence of transgenic animals expressing the chromosomally-integrated fusion protein DAF-16::GFP. These results are presented in supplementary figure 2B. We found that DAF-16 expression levels are not increased following training, but actually decrease by ~25% in the P0-trained animals compared to naive animals. Expression levels then rise back in the F1 progeny.

In the results section we added (page 4, lines 98-101):

“In addition, starvations did not increase expression levels of DAF-16 in the trained parental generation, nor in their F1 progeny, thus precluding the possibility that animals undergoing repeated starvations have a lower threshold for initiation of the stress response due to higher levels of DAF-16 (Supplementary figure 2B).”

While both the trained and mock-trained groups are undergoing the same starvation conditions, it may explain why we see increases of DAF-16 translocation in all groups in supplementary figure 1.

It is very likely that the higher % of P0 animals observed with nuclear DAF-16 before the challenge with IAA (15%) is due to the repeated starvations. This is in contrast to the ~5% nuclear DAF-16 before the challenge found in the subsequent-generation animals, as these animals never experienced starvations nor any other stress.

Nevertheless, while challenging the mock trained animals with IAA did not change the % animals with nuclear DAF-16, the trained animals showed a significant increase to ~50% in the P0 generation (and somewhat lower, yet significantly higher, levels in F1 and F2). Thus, while the P0 trained animals may be more susceptible due to the harsh repeated starvations, only the trained animals show stress induction following memory reactivation.

Minor

1. For figure 1 panels B,C, and D clearer labeling is needed to identify which structures are being identified by the white arrows. Inclusion of a scale bar and anterior/posterior compass would further increase figure clarity.

We have now reproduced the images and provided them in a higher resolution and focusing on the sheath cells to better show the nuclear DAF-16 (denoted by arrowheads). We have also included a scale bar to all panels.

2. For figure 1 panel E, the “F2 no pre-selection” bars should be on the same axis as the others conditions.

This figure (now fig 2A) was entirely reformatted to include the newly added data. We now provide quantification for selected and non-selected F1s, F2s and F3s, and all the bars are found on the same axis.

3. Figure 2C. What is the purpose of including the data for the *hrde-1* KO if the data is uninterpretable?

Since *hrde-1* was shown in several reports as a key player in small RNA inheritance, we thought it was important to include its data in the main text. Furthermore, the new data that scores the behavioral outputs following memory retrieval show that *hrde-1* is involved in this inheritance (P0s avoid the CS IAA while F1s do not, fig 6B). Also, the fact that *hrde-1* mutants are impaired to form the cellular associative memory already at the trained-P0 generation may also be of value to the readers.

REVIEWER COMMENTS

Reviewer #1 (Remarks to the Author):

The authors have substantially improved their manuscript on the inheritance of associative memory in *C. elegans*, by further clarifying the text and by adding the results of additional experimental work. The most important additions to the work are the behavioral data showing the inheritance of associative (behavioral) memory across generations, and several controls necessary to support the authors' claims. This has greatly improved the study and strengthens the conclusions emerging from this work. The underlying molecular and cellular mechanisms were not explored in more detail. Although insights into these processes remain rather general and mainly focused on previously identified pathways, the authors aim to address these mechanisms in more depth in future studies.

The authors have addressed most reviewer comments well and in depth, although some remarks remain to be resolved:

1. Related to comment 2 of reviewer 1: The authors have verified that all mutants used in this study have been backcrossed at least 5 times. However, backcrossing with N2 wild type animals does not remove all background mutations, especially not those that are closely linked to the locus of interest. Previous work, for example, from the Hobert lab has also shown that mutant strains still contain a high mutational load even after several rounds of backcrossing (e.g. Sarin et al., *Genetics*, 2010; PMID 20439776). Without additional genetic evidence (such as an independent mutant allele or rescue experiment), the authors cannot claim that the genes under study are required or involved in memory inheritance; their experiments merely suggest that they play a role. If such experiments cannot be provided (although standard in the field), they should at least acknowledge that effects of possible background mutations in the mutant strains cannot be ruled out, and tone down the conclusions on specific genes being required/involved, being key factors, etc. in memory inheritance.

2. Related to new Figures 5 and 6: Graphs showing the inheritance of cellular and behavioral responses in mutant strains do not include wild type (WT) controls. Therefore (and because memory assays are intrinsically variable), several claims in the text are not well supported: 1) For the cellular response, all mutant strains in panels 5A, 6A and 6C show no or reduced inheritance. Since these graphs do not include data of WT controls (tested in the same experiment as the mutant strains), it is unclear if or how much the mutants are defective, or whether there may have been something wrong in the experimental conditions on those assay days. 2) Similarly, none of the strains in panels 5B and 6D show inheritance of the behavioral avoidance response, and the graphs do not include data showing that WT control animals did inherit avoidance responses in these experiments. Only for NRDE-3, the authors added a note to compare the mutant strain in Figure 6A/B to WT controls in Figures 2A and 4A; however, it is not clear whether these WT controls were tested in parallel with the mutant strain. Data from WT controls tested in the same experiments is necessary to draw solid conclusions on the possible effects of these genes on memory inheritance.

3. Related to new Figure 7: In the revised manuscript, the authors have tested mutants of epigenetic and neuropeptide pathways for the inheritance of both cellular stress and behavioral responses. However, the role of the AWC neuron and the serotonin pathway, established in their previous work, was assessed only for the inheritance of the cellular stress response (Figure 7). It would be good if the authors could extend their investigation of these previously established mechanisms to the inheritance of behavioral responses. If this is not feasible, they should clarify in the results why they investigated the role of these pathways only in the inheritance of cellular memory and distinguish more clearly between the inheritance of both types of responses in the abstract, introduction and results section (as was done in the discussion). Changing the order of describing these experiments in the results could also improve the clarity of the manuscript. For example, by describing the investigation of mechanisms for inheriting the cellular response separately from those for the inheritance of behavioral responses.

4. Related to lines 75-76 and 538-539: The authors have refrained from drawing conclusions on the

fitness advantage of memory inheritance, but include a strong statement at the end of the introduction and discussion, saying that “The inherited memory increased the progeny survival chances, providing the evolutionary basis for the emergence of this valuable capacity in the Animal Kingdom.” They should tone down this statement. Although an increase in progeny survival chances may be one advantage of the inherited memory, they have not shown that it provides the evolutionary basis for the emergence of memory inheritance.

Reviewer #2 (Remarks to the Author):

The revised manuscript is much improved with the additional data, and the reviewers have addressed my previous comments.

I have the following suggestions to improve the revised manuscript even more:

Figure 1 B & C: insert panels of a zoomed in gonadal sheath cell might be helpful. Although these images are much better than those shown in the previous version of the manuscript, it is somewhat difficult to ascertain the nuclei at the current magnification displayed. Additionally, I would like to see the graphs quantifying Figure 1 B&C in this figure. As it is this currently presented, this information is in Supplemental figure 1 and Figure 2, and so the title of Figure 1 “aversive associative memories are transmitted to the F1 and the F2, but not to the F3 generations” is not an accurate reflection of the data shown in the Figure. I think Figure 1 and 2 should be combined.

Figure 2: The claim that transmission of the associative memory is via sperm only is true for the F1 generation but has not been tested for the F2, so it should not be mentioned in the figure title.

Line 154 – use ‘contrast’ instead of ‘contradistinction’.

Figure 5: It would be easier to interpret this figure if the wildtype results were also included on the graph.

Line 333 there should be an additional closed bracket after move and a new open bracket before Figure 6

It is intriguing that the stage-specific training does not result in memory formation with the exception of L4 training. This is the age at which spermatogenic genes are highly expressed during germline development which is intriguing since these memories only seem to be transmitted through sperm. I’m sure that there are other important papers in this space, but one from the Cecere lab (Cornes et al, Dev Cell 2022) came out very recently and discusses small RNAs and their perturbation in similar mutants to those used in this study, particular in L4 animals. I think this is worth including in the discussion., perhaps around Lines 480-489.

Reviewer #3 (Remarks to the Author):

The new version of this manuscript is improved and I think with the minor corrections discussed below it will be ready to publish.

While the corrections in the rebuttal seem appropriate in the actual text intro it still says three main mechanisms may underlie transmittance of epigenetic info (top of page 3)... The authors seem to have corrected this in the main body but not in the introduction. I already stated in my last review that this is inaccurate and ignores the body of literature talking about other modes of inheritance (prions, maternally inherited proteins, microbiota, lipids, rRNAs, tRNA fragments ...) this needs to be corrected before this paper can be published. What was written in the rebuttal would be appropriate (I think it was just forgotten to update in the final text!).

It seems like Figure 2 should be included in Figure 1 since it is the quantitation of the representative images shown in Figure 1.

I’m not clear what “withdrawal” actually is here. The authors cite a paper and assume the readers will read that paper in addition to their own. It would be better to more fully describe what the phenotype is

that is being assessed here in addition to citing the initial paper.

WDR-5.1 is not an H3K4 methyltransferase. It is part of the complex which methylates H3K4 but the actual methylase is either SET-2 or SET-16.

Reviewer #4 (Remarks to the Author):

Deshe et al. have completed an extensive amount of revision experiments in response to the reviewers concerns. The amount of work the authors completed is impressive and this paper adds to the body of literature which shows that stress and starvation can induce behavioral and physiological changes across generations. While the authors do address the majority of the reviewer's questions and concerns in some fashion, some critical issues remain unanswered.

1) While the addition of behavioral data shows there is some transgenerational inheritance of behavioral response (withdrawal) to IAA, it is extremely concerning the authors found that behavioral withdrawal memory can exist in the absence of DAF-16 nuclear translocation and vice versa (Supplemental Fig. 4). This finding affects the entire paper and should be immediately reported early on and not included as a supplemental figure. If authors are using DAF-16 translocation as a readout that "cellular memory" is occurring but translocation and behavioral memory are not correlated suggests that the authors are actually studying two processes that are induced by the starvation training paradigm. If DAF-16 is being used a readout for cellular changes associated with memory, but they do not always correlate or reflect behavioral memory, then using DAF-16 as a proxy is insufficient. The new data indicates that DAF-16 translocation is neither necessary nor sufficient for the behavioral change. DAF-16 translocation can be reported, but should not be called memory. The authors need to re-write the paper reflecting this mismatch between the two processes to reflect that they are studying 2 separate process in response to the training (possibly stress)- the nuclear translocation and the behavioral change, however these are not always correlated. The data is good and interesting, however as the paper is written the interpretation is not correct.

A major concern that is completely ignored is that memory is a behavioral construct, not a physiological processes (Reviewer 4 – major comment 1). Behavioral terms cannot be used for cellular processes. The authors use the term "cellular memory" countless times throughout the manuscript, which is not correct. An accurate term would be "cellular changes" or "nuclear translation" is occurring in response to training or repeated exposure to starvation, not cellular memory of associative memory. A concrete example of the issue being raised is the case where someone falls and breaks their leg and its repair is incomplete, so they still limp. Would we call the limp "a memory of the fall"? We would not. For learning theorists the definition of learning and memory states that learning is a lasting change in behavior as a result of experience, however it also includes several caveats about what learning is not- it is not learning if the changes are due to injury/damage, drugs, developmental processes or change of state.

Further, while the authors attempted to show whether the "cellular memory" could undergo extinction (note- memory is not "extinct" but "undergoes extinction" – used incorrectly throughout manuscript) with repeated exposures to IAA, the authors do not show whether the memory to withdrawal from IAA could undergo extinction (Figure 3). The authors did attempt to address the concerns about memory extinction, however the experiment was conducted using the wrong phenotype (daf-16 translocation rather than behavior) studied.

Lastly, there is a potential typo on line 571. It currently states that there were 4 starvation periods with the final one lasting ~65 hours long. We assume that the authors meant ~6.5 hours, but clarification is needed. If the starvation treatment is multiple long starvation periods this would induce large changes in gene expression and animal physiology which would dramatically change the animal and could interfere with the biological processes studied here including locomotion and neurotransmitter release regulation (<https://www.ncbi.nlm.nih.gov/pmc/articles/PMC4587315/>).

A point by point response to reviewers' comments

We thank the reviewers for the helpful comments and suggestions. We have now addressed all the comments and applied all of the suggestions. Specifically, we performed the required experiments to analyze behavior of mutants in parallel to WT animals, and analyzed if memory evoked avoidance undergoes extinction in both, the trained P0 and their F1 progeny. We modified graph panels as suggested to make them clearer, and importantly, we now carefully followed the recommendation for rephrasing and referring to the odor-evoked DAF-16 translocation as acquired cellular changes (rather than associative memory). We also followed the suggestion to clearly separate the two distinct inherited outputs reported herein.

Reviewer #1 (Remarks to the Author):

The authors have substantially improved their manuscript on the inheritance of associative memory in *C. elegans*, by further clarifying the text and by adding the results of additional experimental work. The most important additions to the work are the behavioral data showing the inheritance of associative (behavioral) memory across generations, and several controls necessary to support the authors' claims. This has greatly improved the study and strengthens the conclusions emerging from this work. The underlying molecular and cellular mechanisms were not explored in more detail. Although insights into these processes remain rather general and mainly focused on previously identified pathways, the authors aim to address these mechanisms in more depth in future studies.

The authors have addressed most reviewer comments well and in depth, although some remarks remain to be resolved:

1. Related to comment 2 of reviewer 1: The authors have verified that all mutants used in this study have been backcrossed at least 5 times. However, backcrossing with N2 wild type animals does not remove all background mutations, especially not those that are closely linked to the locus of interest. Previous work, for example, from the Hobert lab has also shown that mutant strains still contain a high mutational load even after several rounds of backcrossing (e.g. Sarin et al., *Genetics*, 2010; PMID 20439776). Without additional genetic evidence (such as an independent mutant allele or rescue experiment), the authors cannot claim that the genes under study are required or involved in memory inheritance; their experiments merely suggest that they play a role. If such experiments cannot be provided (although standard in the field), they should at least acknowledge that effects of possible background mutations in the mutant strains cannot be ruled out, and tone down the conclusions on specific genes being required/involved, being key factors, etc. in memory inheritance.

We agree, and we have now toned down the conclusions stating that these genes may merely be involved in the inheritance. In the results section where we summarize the findings, we state (lines 387-389):

“Together, these results suggest that H3K9 and H3K36 methylation, the Argonaute NRDE-3, and neuropeptide signaling may be involved in inheritance of the acquired cellular changes.”

Also, in the discussion, we provided the following explanation (lines 546-549):

“Nevertheless, our experiments using the various mutant strains (histone modifiers, small RNA-related genes and neuro-transmitter/peptide secretion factors) provide merely correlative, rather than causative, indication for their possible involvement in both kinds of inheritances”.

2. Related to new Figures 5 and 6: Graphs showing the inheritance of cellular and behavioral responses in mutant strains do not include wild type (WT) controls. Therefore (and because memory assays are intrinsically variable), several claims in the text are not well supported: 1) For the cellular response, all mutant strains in panels 5A, 6A and 6C show no or reduced inheritance. Since these graphs do not include data of WT controls (tested in the same experiment as the mutant strains), it is unclear if or how much the mutants are defective, or whether there may have been something wrong in the experimental conditions on those assay days. 2) Similarly, none of the strains in panels 5B and 6D show inheritance of the behavioral avoidance response, and the graphs do not include data showing that WT control animals did inherit avoidance responses in these experiments. Only for NRDE-3, the authors added a note to compare the mutant strain in Figure 6A/B to WT controls in Figures 2A and 4A; however, it is not clear whether these WT controls were tested in parallel with the mutant strain. Data from WT controls tested in the same experiments is necessary to draw solid conclusions on the possible effects of these genes on memory inheritance.

In the experiments assaying cellular changes (DAF-16 nuclear translocation), we always assayed WT animals alongside the mutants. We now provide these WT results alongside the mutants in the revised figure 6A. In figure 3C, in which we first report the cellular change phenomenon, we grouped all the WT experiments, including those performed in parallel to the mutants.

Regarding the second point concerning inheritance of the avoidance behavior: Most of the mutants were assayed in parallel to the WT animals. For some of the mutants that were not assayed alongside the WT, we reassayed them together with WT. We now provide all the experimental results of WT and mutants in the same figure (revised figure 2).

3. Related to new Figure 7: In the revised manuscript, the authors have tested mutants of epigenetic and neuropeptide pathways for the inheritance of both cellular stress and behavioral responses. However, the role of the AWC neuron and the serotonin pathway, established in their previous work, was assessed only for the inheritance of the cellular stress response (Figure 7). It would be good if the authors could extend their investigation of these previously established mechanisms to the inheritance of behavioral responses. If this is not feasible, they should clarify in the results why they investigated the role of these pathways only in the inheritance of cellular memory and distinguish more clearly between the inheritance of both types of responses in the

abstract, introduction and results section (as was done in the discussion). Changing the order of describing these experiments in the results could also improve the clarity of the manuscript. For example, by describing the investigation of mechanisms for inheriting the cellular response separately from those for the inheritance of behavioral responses.

We were more excited and interested to study inheritance of cellular changes rather than that of behavioral outputs (which were documented before). This is the reason we focused the efforts on this phenomenon. We have closely followed the recommendations and we now clearly segregate between the two inheritance forms. This is reflected throughout the paper: Already in the title, we refer to the two inherited forms, and in the abstract, we describe them separately. In the results section, we begin with the behavioral outputs (figs 1-2) and then switch to describe the cellular changes (fig 3-8) where in fig. 3 we actually analyze if both forms are functionally coupled. Similarly, in the discussion, we discuss each of the inherited forms separately and then compare and contrast them in light of our new data showing that these processes are probably uncoupled.

4. Related to lines 75-76 and 538-539: The authors have refrained from drawing conclusions on the fitness advantage of memory inheritance, but include a strong statement at the end of the introduction and discussion, saying that “The inherited memory increased the progeny survival chances, providing the evolutionary basis for the emergence of this valuable capacity in the Animal Kingdom.” They should tone down this statement. Although an increase in progeny survival chances may be one advantage of the inherited memory, they have not shown that it provides the evolutionary basis for the emergence of memory inheritance.

Indeed, and we have now toned down the statements:

We completely removed the statement from the end of the introduction and in the discussion we changed to (line 624):

“In summary, here we established that associative memories and acquired cellular changes are heritable and that they provide the progeny valuable behavioral and cellular/physiological protective instructions that ultimately increase their survival chances”.

Reviewer #2 (Remarks to the Author):

The revised manuscript is much improved with the additional data, and the reviewers have addressed my previous comments.

I have the following suggestions to improve the revised manuscript even more:

Figure 1 B & C: insert panels of a zoomed in gonadal sheath cell might be helpful. Although these images are much better than those shown in the previous version of the manuscript, it is somewhat difficult to ascertain the nuclei at the current magnification displayed. Additionally, I would like to see the graphs quantifying Figure 1 B&C in this figure. As it is this currently presented, this information is in Supplemental figure 1 and Figure 2, and so the title of Figure 1 “aversive associative memories are transmitted to the F1 and the F2, but not to the F3

generations” is not an accurate reflection of the data shown in the Figure. I think Figure 1 and 2 should be combined.

We have now added a zoom-in panel that shows the accumulation of DAF-16 in nuclei of gonad sheath cells (now in figure 3A). The graphs quantifying the nuclear translocation in both P0 and F1s also appear in the same figure (fig. 3C). Thus, we combined all the panels to the same figure and provided the matching title:

“Figure 3. Acquired cellular changes are transmitted up to the F2 generation, and are presumably decoupled from the memory-evoked avoidance behavior.”

Figure 2: The claim that transmission of the associative memory is via sperm only is true for the F1 generation but has not been tested for the F2, so it should not be mentioned in the figure title.

We have now changed the text to refer to F1 only. This appears in figure 7 in the revised version:

“Figure 7. The cellular changes are transmitted to the F1 descendants via the sperm and not the oocytes.”

Line 154 – use ‘contrast’ instead of ‘contradistinction’.

We have changed accordingly.

Figure 5: It would be easier to interpret this figure if the wildtype results were also included on the graph.

We have added the wt data to the graph. In the revised version, it is shown in figures 6.

Line 333 there should be an additional closed bracket after move and a new open bracket before Figure 6

The text has been revised.

It is intriguing that the stage-specific training does not result in memory formation with the exception of L4 training. This is the age at which spermatogenic genes are highly expressed during germline development which is intriguing since these memories only seem to be transmitted through sperm. I’m sure that there are other important papers in this space, but one from the Cecere lab (Cornes et al, Dev Cell 2022) came out very recently and discusses small RNAs and their perturbation in similar mutants to those used in this study, particular in L4 animals. I think this is worth including in the discussion., perhaps around Lines 480-489.

We now added this point to the discussion and included the relevant study by Cornes et al (lines 568-577):

*“Though the germline starts to develop already at the second larval stage, a single training cycle during either the L1, L3, or the L4 larval stages did not lead to inheritance of the cellular changes (**Supplementary figure 3**). Spermatogenesis initiates during the L4 larval stage and is accompanied by substantial transcriptional changes, including CSR-1-mediated licensing and HRDE-1-assisted piRNA-triggered silencing of hundreds of genes, that are required for sperm differentiation and proper function (Cornes et al., 2022). Yet, while a single training at the L4 stage induced the cellular changes at the parental generation, this information failed to be transmitted to the progeny, indicating that repeated training cycles are crucial for a potent information transmission“.*

Reviewer #3 (Remarks to the Author):

The new version of this manuscript is improved and I think with the minor corrections discussed below it will be ready to publish.

While the corrections in the rebuttal seem appropriate in the actual text into it still says three main mechanisms may underlie transmittance of epigenetic info (top of page 3)... The authors seem to have corrected this in the main body but not in the introduction. I already stated in my last review that this is inaccurate and ignores the body of literature talking about other modes of inheritance (prions, maternally inherited proteins, microbiota, lipids, rRNAs, tRNA fragments ...) this needs to be corrected before this paper can be published. What was written in the rebuttal would be appropriate (I think it was just forgotten to update in the final text!).

We have now corrected this sentence in the introduction (lines 57-60):

“...their compatibility with a myriad of genetic manipulations already extracted evolutionarily conserved mechanisms that promote epigenetic inheritance, including neuropeptides, proteins, chromatin modifications, small RNAs and other factors...“

It seems like Figure 2 should be included in Figure 1 since it is the quantitation of the representative images shown in Figure 1.

We have now combined figures 1 and 2 into the new figure 3.

I'm not clear what "withdrawal" actually is here. The authors cite a paper and assume the readers will read that paper in addition to their own. It would be better to more fully describe what the phenotype is that is being assessed here in addition to citing the initial paper.

For consistency throughout the manuscript, we changed the phrasing from 'withdrawal' to 'avoiding'. We also added to the Methods, under the 'Avoidance assays' section, how we defined avoidance (lines 718-720):

"Worms that stopped in front of the IAA and backed within three seconds were scored as avoiding; worms that only briefly stopped, or continued crawling forward to cross the IAA stripe were scored as non-avoiding."

WDR-5.1 is not an H3K4 methyltransferase. It is part of the complex which methylates H3K4 but the actual methylase is either SET-2 or SET-16.

We have now corrected accordingly (line 159):

"...WDR-5.1, part of the H3K4 methylation complex, ..."

Reviewer #4 (Remarks to the Author):

Deshe et al. have completed an extensive amount of revision experiments in response to the reviewers concerns. The amount of work the authors completed is impressive and this paper adds to the body of literature which shows that stress and starvation can induce behavioral and physiological changes across generations. While the authors do address the majority of the reviewer's questions and concerns in some fashion, some critical issues remain unanswered.

1) While the addition of behavioral data shows there is some transgenerational inheritance of behavioral response (withdrawal) to IAA, it is extremely concerning the authors found that behavioral withdrawal memory can exist in the absence of DAF-16 nuclear translocation and vice versa (Supplemental Fig. 4). This finding affects the entire paper and should be immediately reported early on and not included as a supplemental figure.

If authors are using DAF-16 translocation as a readout that "cellular memory" is occurring but translocation and behavioral memory are not correlated suggests that the authors are actually studying two processes that are induced by the starvation training paradigm. If DAF-16 is being used a readout for cellular changes associated with memory, but they do not always correlate or reflect behavioral memory, then using DAF-16 as a proxy is insufficient. The new data indicates that DAF-16 translocation is neither necessary nor sufficient for the behavioral change.

We now modified the text to report this finding in the main text (Figure 3D-F), and included a thorough analysis which shows that the two processes are likely to be uncoupled. Thus, behavioral withdrawal memory can exist in the absence of DAF-16 nuclear translocation and vice versa. The revised version now presents these two training-induced outputs as distinct independent processes. As such, we do not claim that DAF-16 is a measure (or a proxy) for the behavioral output or vice versa.

DAF-16 translocation can be reported, but should not be called memory. The authors need to re-write the paper reflecting this mismatch between the two processes to reflect that they are studying 2 separate processes in response to the training (possibly stress)- the nuclear translocation and the behavioral change, however these are not always correlated. The data is good and interesting, however as the paper is written the interpretation is not correct.

We have now rewritten the manuscript as suggested and clearly reflect the mismatch between the two processes (as described above). We also refrain from referring to DAF-16 translocation as a memory, and we strictly use the suggested term 'acquired cellular changes'.

A major concern that is completely ignored is that memory is a behavioral construct, not a physiological process (Reviewer 4 – major comment 1). Behavioral terms cannot be used for cellular processes. The authors use the term “cellular memory” countless times throughout the manuscript, which is not correct. An accurate term would be “cellular changes” or “nuclear translation” is occurring in response to training or repeated exposure to starvation, not cellular memory of associative memory. A concrete example of the issue being raised is the case where someone falls and breaks their leg and its repair is incomplete, so they still limp. Would we call the limp “a memory of the fall”? We would not. For learning theorists the definition of learning and memory states that learning is a lasting change in behavior as a result of experience, however it also includes several caveats about what learning is not- it is not learning if the changes are due to injury/damage, drugs, developmental processes or change of state.

As suggested, we have now rephrased the terminology and refer to it as “inherited cellular changes”. We wish to emphasize that these cellular changes are the result of coupling between the odorant and starvation: DAF-16 nuclear translocation occurs only in response to a re-exposure to the training odorant. Mock-trained animals that underwent the same starvations (in the absence of the odorant) do not show DAF-16 nuclear translocation upon re-exposure to the odorant. Thus, as long as the trained animals are not exposed to the odorant used during training, there is no ‘phenotype’ to the past experience.

Further, while the authors attempted to show whether the “cellular memory” could undergo extinction (note- memory is not “extinct” but “undergoes extinction” – used incorrectly throughout manuscript) with repeated exposures to IAA, the authors do not show whether the memory to withdrawal from IAA could undergo extinction (Figure 3). The authors did attempt to address the concerns about memory extinction, however the experiment was conducted using the wrong phenotype (daf-16 translocation rather than behavior) studied.

We have now performed the requested experiments and studied whether the behavioral phenotype can also undergo extinction. We analyzed both the trained P0 generations as well as their F1 progeny following repeated rounds of exposing the animals to the odorant in the presence of food. The new data is now provided in figure 1C. Similar to the acquired cellular memory, the odor-evoked avoidance output was rather stable and went extinct only after three recovery cycles in the P0 generation. Memory traces were still evident in the F1 generation,

despite repeated extinction attempts in the F1 generation (and unlike the acquired cellular changes, figure 4).

We have also corrected the terminology indicating a memory that undergoes extinction (rather than 'memory extinct').

Lastly, there is a potential typo on line 571. It currently states that there were 4 starvation periods with the final one lasting ~65 hours long. We assume that the authors meant ~6.5 hours, but clarification is needed. If the starvation treatment is multiple long starvation periods this would induce large changes in gene expression and animal physiology which would dramatically change the animal and could interfere with the biological processes studied here including locomotion and neurotransmitter release regulation (<https://www.ncbi.nlm.nih.gov/pmc/articles/PMC4587315/>).

We note that in all experiments, we always assayed and compared trained (starved + odorant) and mock trained (starved only) groups. This is not a typo, and we are aware that significant gene expression and physiological changes occur during this starvation time period in both trained and mock-trained animals. However, we find stark significant differences between the two groups, where re-exposure to the odorant induces DAF-16 translocation and avoidance in trained animals while no such outputs are observed in the mock-trained animals upon re-exposure to the training odorant. Thus the odor-induced behavioral outputs and the acquired changes occur on top of the training (starvation)-induced physiological changes.

REVIEWER COMMENTS

Reviewer #1 (Remarks to the Author):

The authors have addressed my concerns. They have performed several additional experiments, included controls, and clarified the main text and figures to address the reviewers' comments. The additional experiments, analyses, and discussion have substantially improved the quality of the manuscript, which I recommend for publication.

Reviewer #2 (Remarks to the Author):

The authors have substantially re-written the manuscript and as part of the restructure have addressed all my previous comments satisfactorily. I do have a couple of suggestions/queries outlined below and some minor edits, including the addition of some key references that it would be good to cite regards the histone modifiers and small RNAs. They have not investigated the underlying molecular mechanisms in any more detail than previously. This would strengthen the study.

Line 98-99: The first sentence of this paragraph is poorly worded and difficult to understand. I would instead write "When assaying all F1 progeny, we could not detect an elevated avoidance response upon exposure to IAA (Figure 1B). However, when..."

Line 152 A couple of key references missing here for set-25, including Lev et al 2019 and Woodhouse et al 2018.

Line 166-167 The authors have forgotten to cite Shirayama et al 2012 and Ashe et al 2012, which were published at a similar time to Buckley et al 2012 and reported similar data.

Figure 2A was figure 5B in the previous version. The data appear to be identical, except that the wildtype comparisons have been added at the request of previous reviewers. However, the statistical results presented now differ, although the test performed (proportion test) remains the same. The conclusions drawn remain the same, except for hpl-2 which has changed from a NS different between trained and mock trained in the F1 to an apparently highly significant (***) change. This change in statistical outcome has been reflected in the text:

Current version (line 161): the heterochromatin protein HPL-2, which binds methylated histones, do not seem to play a major role in memory inheritance as F1 descendants show memory retrieval capacities

Previous version (line 291): In contrast, inheritance of the behavioral memory was impaired in both of these mutants, suggesting that these genes play a role in mediating this type of memory to the progeny

This is problematic as the raw data does not appear to have changed, so I am curious how the data can go from NS to *** when other tests in the same graph have remained the same, and recommend the authors check their analysis.

Line 320: the authors say that they previously showed that an odor-induced physiological response increases subsequent survival chances when exposed to a stress (in this case heat). Now, they want to test whether inherited cellular changes (which they are measuring by DAF-16 nuclear localisation) also leads to this protective effect in the next generation. I do not see how the experiment described in lines 323-336 tests this. I agree that it tests whether the protective effect can be observed in the F1 generation, but it is not clear that this is necessarily associated with the cellular change in DAF-2 localisation. Figure 4 shows that ~20% of trained F1 animals have nuclear DAF-16 and

Figure 5 shows a relatively subtle increase in F1 survival after heat stress (~5% increase in most pairs), but unless the P0 animals were selected to be those with nuclear DAF-16 (similar to the selection of those that avoided the IAA in early figures), then the authors need to modify their claims/wording, particularly in line 320.

Line 496-7 add odor before avoidance – ‘the first is evidence by the odor avoidance of...’

Line 561 remove comma after both

Reviewer #3 (Remarks to the Author):

The authors have addressed all my concerns and the manuscript should be published. In the last paragraph of the discussion the authors might want to also include a reference to Wang SY et al Cell Rep 2022 which showed a paternal inheritance with small RNAs in response to hypoxia and seems like it would provide additional support beyond Toker et al 2022.

Reviewer #4 (Remarks to the Author):

The authors have done a great amount of work in response to the reviewers' comments, including additional experiments and analyses, and content revisions. The authors have satisfactorily addressed our previous comments. We particularly appreciate the new analysis to show that the inherited behavioral response and inherited cellular response appeared to be uncoupled, which potentially leads to more research questions for the future. The terminology changes in the writing also comply with the consensus in the field of behavioral and cellular studies. We have no further comments on this manuscript and would like to congratulate the authors for completing a nice piece of work.

A point-by-point response to the reviewers' comments

Reviewer #1 (Remarks to the Author):

The authors have addressed my concerns. They have performed several additional experiments, included controls, and clarified the main text and figures to address the reviewers' comments. The additional experiments, analyses, and discussion have substantially improved the quality of the manuscript, which I recommend for publication.

Thank you.

Reviewer #2 (Remarks to the Author):

The authors have substantially re-written the manuscript and as part of the restructure have addressed all my previous comments satisfactorily. I have do have a couple of suggestions/queries outlined below and some minor edits, including the addition of some key references that it would be good to cite regards the histone modifiers and small RNAs. They have not investigated the underlying molecular mechanisms in any more detail than previously. This would strengthen the study.

Line 98-99: The first sentence of this paragraph is poorly worded and difficult to understand. I would instead write "When assaying all F1 progeny, we could not detect an elevated avoidance response upon exposure to IAA (Figure 1B). However, when..."

Thank you. We have now changed as suggested.

Line 152 A couple of key references missing here for set-25, including Lev et al 2019 and Woodhouse et al 2018.

Indeed, and we have now added these refs.

Line 166-167 The authors have forgotten to cite Shirayama et al 2012 and Ashe et al 2012, which were published at a similar time to Buckley et al 2012 and reported similar data.

Thank you for pointing this out. We have now added these refs.

Figure 2A was figure 5B in the previous version. The data appear to be identical, except that the wildtype comparisons have been added at the request of previous reviewers. However, the statistical results presented now differ, although the test performed (proportion test) remains the same. The conclusions drawn remain the same, except for hpl-2 which has changed from a NS different between trained and mock trained in the F1 to an apparently highly significant (***) change. This change in statistical outcome has been reflected in the text:

Current version (line 161): the heterochromatin protein HPL-2, which binds methylated histones, do not seem to play a major role in memory inheritance as F1 descendants show memory retrieval capacities

Previous version (line 291): In contrast, inheritance of the behavioral memory was impaired in both of these mutants, suggesting that these genes play a role in mediating this type of memory to the progeny

This is problematic as the raw data does not appear to have changed, so I am curious how the data can go from NS to *** when other tests in the same graph have remained the same, and recommend the authors check their analysis.

Following reviewer's request, we added 3 new experimental repeats for the hpl-2 mutant. As a result, the data size increased from 3 data points to 6 data points. This increase in sample size changed the statistical results which now became significant: In the previous version, the p-value was **0.06**, while now the p-value is **0.0019**. Thus, as the trained animals are significantly different from the mock trained animals (in both P0 and F1), we concluded that this gene does not play a role in the associative memory nor in its inheritance. We did have a mistake marking the significance with *** while it should be **. We now provide the same figure with the correct p-values asterisks.

Line 320: the authors say that they previously showed that an odor-induced physiological response increases subsequent survival chances when exposed to a stress (in this case heat). Now, they want to test whether inherited cellular changes (which they are measuring by DAF-16 nuclear localisation) also leads to this protective effect in the next generation. I do not see how the experiment described in lines 323-336 tests this. I agree that it tests whether the protective effect can be observed in the F1 generation, but it is not clear that this is necessarily associated with the cellular change in DAF-2 localisation. Figure 4 shows that ~20% of trained F1 animals have nuclear DAF-16 and Figure 5 shows a relatively subtle increase in F1 survival after heat stress (~5% increase in most pairs), but unless the P0 animals were selected to be those with nuclear DAF-16 (similar to the selection of those that avoided the IAA in early figures), then the authors need to modify their claims/wording, particularly in line 320.

We have reworded the text to be more cautious about the claims made. We now merely suggest that this may be due to DAF-16 translocation:

In the results, line 342-3:

".... presumably due to odor-induced DAF-16 nuclear translocation".

And in the discussion, line 518-9:

"... presumably due to odor-induced DAF-16 nuclear translocation (Figure 5)."

Line 496-7 add odor before avoidance – 'the first is evidence by the odor avoidance of...'
Added.

Line 561 remove comma after both
Removed.

Reviewer #3 (Remarks to the Author):

The authors have addressed all my concerns and the manuscript should be published. In the last paragraph of the discussion the authors might want to also include a reference to Wang SY et al Cell Rep 2022 which showed a paternal inheritance with small RNAs in response to hypoxia and seems like it would provide additional support beyond Toker et al 2022.

Thank you for referring us to this ref. We have now added it to the text (line 619).

Reviewer #4 (Remarks to the Author):

The authors have done a great amount of work in response to the reviewers' comments, including additional experiments and analyses, and content revisions. The authors have satisfactorily addressed our previous comments. We particularly appreciate the new analysis to show that the inherited behavioral response and inherited cellular response appeared to be uncoupled, which potentially leads to more research questions for the future. The terminology changes in the writing also comply with the consensus in the field of behavioral and cellular studies. We have no further comments on this manuscript and would like to congratulate the authors for completing a nice piece of work.

Thank you.

REVIEWERS' COMMENTS

Reviewer #2 (Remarks to the Author):

The authors have now addressed all my previous comments and concerns, and I recommend to proceed to publication. Congratulations on preparing and analysing a large amount of extra data and producing a nice paper.

A point-by-point response to the reviewers' comments

Reviewer #2 (Remarks to the Author):

The authors have now addressed all my previous comments and concerns, and I recommend to proceed to publication. Congratulations on preparing and analysing a large amount of extra data and producing a nice paper.

Thank you.